# SCALE-UP: An Efficient Black-box Input-level Backdoor Detection via Analyzing Scaled Prediction Consistency

**Junfeng Guo**[1][*][†]**, Yiming Li**[2][*]**, Xun Chen**[3][‡]**, Hanqing Guo**[4][†]**, Lichao Sun**[5]**, Cong Liu**[6]

[1]Department of Computer Science, UT Dallas
[2]Tsinghua Shenzhen International Graduate School, Tsinghua University
[3]Samsung Research America, Mountain View
[4]Department of Computer Science, Michigan State University
[5]Department of Computer Science, Lehigh University
[6]Department of Electricity and Computer Engineering Department, UC Riverside

## Abstract

Deep neural networks (DNNs) are vulnerable to backdoor attacks, where adversaries embed a hidden backdoor trigger during the training process for malicious prediction manipulation. These attacks pose great threats to the applications of DNNs under the real-world machine learning as a service (MLaaS) setting, where the deployed model is fully black-box while the users can only query and obtain its predictions. Currently, there are many existing defenses to reduce backdoor threats. However, almost all of them cannot be adopted in MLaaS scenarios since they require getting access to or even modifying the suspicious models. In this paper, we propose a simple yet effective black-box input-level backdoor detection, called SCALE-UP, which requires only the predicted labels to alleviate this problem. Specifically, we identify and filter malicious testing samples by analyzing their prediction consistency during the pixel-wise amplification process. Our defense is motivated by an intriguing observation (dubbed *scaled prediction consistency*) that the predictions of poisoned samples are significantly more consistent compared to those of benign ones when amplifying all pixel values. Besides, we also provide theoretical foundations to explain this phenomenon. Extensive experiments are conducted on benchmark datasets, verifying the effectiveness and efficiency of our defense and its resistance to potential adaptive attacks. Our codes are available at https://github.com/JunfengGo/SCALE-UP.

## 1 Introduction

Deep neural networks (DNNs) have been deployed in a wide range of mission-critical applications, such as autonomous driving (Kong et al., 2020; Grigorescu et al., 2020; Wen & Jo, 2022), face recognition (Tang & Li, 2004; Li et al., 2015; Yang et al., 2021), and object detection (Zhao et al., 2019; Zou et al., 2019; Wang et al., 2021). In general, training state-of-the-art DNNs usually requires extensive computational resources and training samples. Accordingly, in real-world applications, developers and users may directly exploit third-party pre-trained DNNs instead of training their new models. This is what we called machine learning as a service (MLaaS).

However, recent studies (Gu et al., 2019; Goldblum et al., 2022; Li et al., 2022a) revealed that DNNs can be compromised by embedding adversary-specified hidden backdoors during the training process, posing threatening security risks to MLaaS. The adversaries can activate embedded backdoors in the attacked models to maliciously manipulate their predictions whenever the pre-defined trigger pattern appears. Users are hard to identify these attacks under the MLaaS setting since attacked DNNs behave normally on benign samples.

---

[*]The first two authors contributed equally to this paper.
[†]This work was done when Junfeng Guo and Hanqing Guo interned in Samsung Research America.
[‡]Corresponding Author: Xun Chen (e-mail: xun.chen@samsung.com).

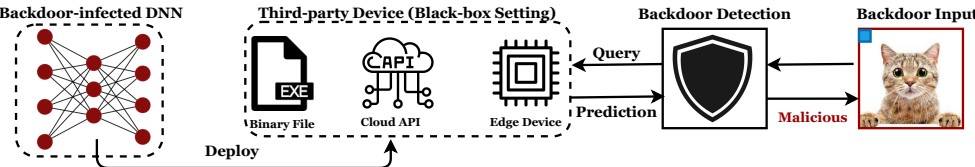

Figure 1: An illustration of the black-box input-level backdoor detection.

To reduce backdoor threats, there are many different types of defenses, such as model repairing (Li et al., 2021b; Wu & Wang, 2021; Zeng et al., 2022), poison suppression (Du et al., 2020; Li et al., 2021a; Huang et al., 2022), and backdoor detection (Xiang et al., 2022; Liu et al., 2022; Guo et al., 2022c;d). However, most of these defenses were designed under the white-box setting, requiring accessing or even modifying model weights. Accordingly, users cannot adopt them in MLaaS scenarios. Currently, there are also a few model-level (Huang et al., 2020; Dong et al., 2021; Guo et al., 2022c) and input-level (Li et al., 2021c; Qiu et al., 2021; Gao et al., 2021) black-box backdoor defenses where users can only access final predictions. However, these defenses have some implicit assumptions of backdoor triggers (*e.g.*, a small static patch), leading to being easily bypassed by advanced backdoor attacks (Nguyen & Tran, 2021; Li et al., 2021d). Their failures raise an intriguing question: *what are the fundamental differences between poisoned and benign samples that can be exploited to design universal black-box backdoor detection?*

In this paper, we focus on the *black-box input-level backdoor detection*, where we intend to identify whether a given suspicious input is malicious based on predictions of the deployed model (as shown in Fig. 1). This detection is practical in many real-world applications since it can serve as the 'firewall' helping to block and trace back malicious samples in MLaaS scenarios. However, this problem is challenging since defenders have limited model information and no prior knowledge of the attack. Specifically, we first explore the pixel-wise amplification effects on benign and poisoned samples, motivated by the understanding that increasing trigger values does not hinder or even improve the attack success rate of attacked models (as preliminarily suggested in (Li et al., 2021c)). We demonstrate that the predictions of attacked images generated by both classical and advanced attacks are significantly more consistent compared to those of benign ones when amplifying all pixel values. We refer to this intriguing phenomenon as *scaled prediction consistency*. In particular, we also provide theoretical insights to explain this phenomenon. After that, based on these findings, we propose a simple yet effective method, dubbed scaled prediction consistency analysis (SCALE-UP), under both *data-free* and *data-limited* settings. Specifically, under the data-free setting, the SCALE-UP examines each suspicious sample by measuring its *scaled prediction consistency* (SPC) value, which is the proportion of labels of scaled images that are consistent with that of the input image. The larger the SPC value, the more likely this input is malicious. Under the data-limited setting, we assume that defenders have a few benign samples from each class, based on which we can reduce the side effects of class differences to further improve our SCALE-UP.

In conclusion, our main contributions are four-fold. **1)** We reveal an intriguing phenomenon (*i.e.*, scaled prediction consistency) that the predictions of attacked images are significantly more consistent compared to those of benign ones when amplifying all pixel values. **2)** We provide theoretical insights trying to explain the phenomenon of scaled prediction consistency. **3)** Based on our findings, we propose a simple yet effective black-box input-level backdoor detection (dubbed 'SCALE-UP') under both data-free and data-limited settings. **4)** We conduct extensive experiments on benchmark datasets, verifying the effectiveness of our method and its resistance to potential adaptive attacks.

## 2 RELATED WORK

### 2.1 BACKDOOR ATTACK

Backdoor attacks (Gu et al., 2019; Li et al., 2022a; Hayase & Oh, 2023) compromise DNNs by contaminating the training process through injecting poisoned samples. These samples are crafted by adding adversary-specified trigger patterns into the selected benign samples. Backdoor attacks are stealthy since the attacked models behave normally on benign samples and the adversaries only need to craft a few poisoned samples. Accordingly, they introduce serious risks to DNN-based applications. In general, existing attacks can be roughly divided into two main categories based on the trigger property, including **1)** patch-based attacks and **2)** non-patch-based attacks, as follows:

**Patch-based Backdoor Attacks.** (Gu et al., 2019) proposed the first backdoor attack, which was called BadNets. Specifically, BadNets randomly selected and modified a few benign training samples by stamping the trigger patch and changing their label with a pre-defined target label. The generated poisoned samples associated with the remaining benign samples will be released to users to train their models. After that, (Chen et al., 2017) first discussed the attack stealthiness and introduced trigger transparency, where they suggested that poisoned images should be indistinguishable compared with their benign version to evade human inspection. (Turner et al., 2019) argued that making trigger patches invisible is not stealthy enough since the ground-truth labels of poisoned samples are different from the target label. They designed the first clean-label backdoor attack where adversaries can only poison samples from the target class. Recently, (Li et al., 2021c) proposed the first physical backdoor attack, where the location and appearance of the trigger contained in the digitized test samples may be different from that of the one used for training.

**Non-patch-based Backdoor Attacks.** Different from classical attacks whose trigger pattern is a small patch, recent advanced methods exploited non-patch-based triggers trying to bypass backdoor defenses. For example, (Zhao et al., 2020a) exploited full-image size targeted universal adversarial perturbation (Moosavi-Dezfooli et al., 2017) as the trigger pattern to design a more effective clean-label backdoor attack. (Nguyen & Tran, 2021) adopted image warping as the backdoor trigger, which deforms the whole image while preserving image content. Recently, (Li et al., 2021d) proposed the first poison-only sample-specific trigger patterns, inspired by the DNN-based image steganography (Tancik et al., 2020). This attack broke the fundamental assumption (*i.e.*, the trigger is sample-agnostic) of most existing defenses and therefore could easily bypass them.

## 2.2 BACKDOOR DEFENSE

Currently, there are many backdoor defenses to alleviate backdoor threats. In general, existing methods can be roughly divided into two main categories based on the defender's capacities, including **1)** white-box defenses and **2)** black-box defenses, as follows:

**White-box Backdoor Defenses.** In these approaches, defenders need to obtain the source files of suspicious models. The most typical white-box defenses are model repairing, aiming at removing hidden backdoors in the attacked DNNs. For example, (Liu et al., 2018a; Wu & Wang, 2021) proposed to remove backdoors based on model pruning; (Li et al., 2021b; Xia et al., 2022) adopted model distillation to eliminate hidden backdoors. There are also other types of white-box defenses, such as poison suppression (Du et al., 2020; Li et al., 2021a; Huang et al., 2022) and trigger reversion (Wang et al., 2019; Hu et al., 2022; Tao et al., 2022). However, users cannot use them under the machine learning as a service (MLaaS) setting, where they can only obtain model predictions.

**Black-box Backdoor Defenses.** In these methods, defenders can only query the (deployed) model and obtain its predictions. Currently, there are two main types of black-box defenses, including **1)** model-level defenses (Huang et al., 2020; Dong et al., 2021; Guo et al., 2022c) and **2)** input-level defenses (Li et al., 2021c; Qiu et al., 2021; Gao et al., 2021). Specifically, the former ones intend to identify whether the (deployed) suspicious model is attacked while the latter ones detect whether a given suspicious input is malicious. In this paper, we focus on the input-level black-box defense since it can serve as the 'firewall' helping to block and trace back malicious samples in MLaaS scenarios. However, as we will demonstrate in our experiments, existing input-level defenses can be easily bypassed by advanced backdoor attacks since they have some strong implicit assumptions of backdoor triggers. How to design effective black-box input-level backdoor detectors is still an important open question and worth further exploration.

## 3 THE PHENOMENON OF SCALED PREDICTION CONSISTENCY

In this section, we explore the prediction behaviors of benign and poisoned samples generated by attacked DNNs since it is the cornerstone for designing black-box input-level backdoor defense. Before illustrating our key observations, we first review the general process of backdoor attacks.

**The Main Pipeline of Backdoor Attacks.** Let $\mathcal{D} = \{(\boldsymbol{x}_i, y_i)\}_{i=1}^{N}$ represent a unmodified benign training set and $C : \mathcal{X} \to \mathcal{Y}$ is the deployed DNN, where $\boldsymbol{x}_i \in \mathcal{X} = [0, 1]^{C \times W \times H}$ is the image, $y_i \in \mathcal{Y} = \{1, \ldots, K\}$ is its label, and $K$ is the number of different labels. The backdoor adversaries will select some benign samples (*i.e.*, $\mathcal{D}_s$) to generate their modified version

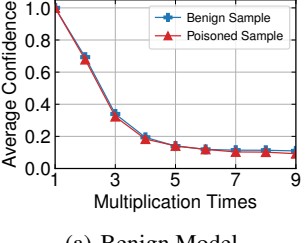
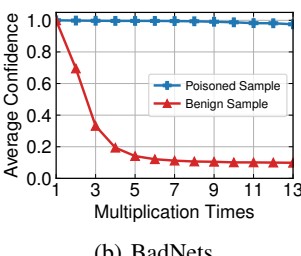
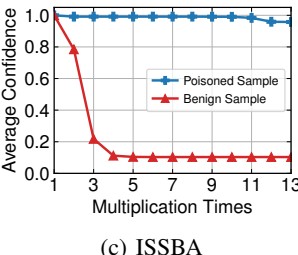

| (a) Benign Model | (b) BadNets | (c) ISSBA |

Figure 2: The average confidence (*i.e.*, average probabilities on the originally predicted label) of benign and poisoned samples *w.r.t.* pixel-wise multiplications under benign and attacked models.

by $\mathcal{D}_m = \{(\boldsymbol{x}', y_t) | \boldsymbol{x}' = \boldsymbol{x} + g(\boldsymbol{x}), (\boldsymbol{x}, y) \in \mathcal{D}_s\}$, where $y_t$ is an adversary-specified target label and $g(\cdot)$ is a pre-defined poison generator. For example, $g(\boldsymbol{x}) = \boldsymbol{m} \odot (\boldsymbol{t} - \boldsymbol{x})$ in BadNets (Gu et al., 2019) and blended attack (Chen et al., 2017), where $\odot$ represents the element-wise product, $m \in [0, 1]^{C \times W \times H}$ is a transparency mask, and $\boldsymbol{t} \in \mathcal{X}$ is the trigger pattern. Given $N_b$ benign samples and $N_p$ poisoned samples, the backdoor adversaries will train the attacked DNN $f(\cdot; \boldsymbol{\theta})$ based on the following optimization process (with loss $\mathcal{L}$):

$$\min_{\boldsymbol{\theta}} \sum_{i=1}^{N_b} \mathcal{L}(f(\boldsymbol{x}_i; \boldsymbol{\theta}), y_i) + \sum_{j=1}^{N_p} \mathcal{L}(f(\boldsymbol{x}'_j; \boldsymbol{\theta}), y_t). \tag{1}$$

As preliminarily demonstrated in (Li et al., 2021c), increasing the pixel value of backdoor triggers does not hinder or even improve the attack success rate. However, defenders can not accurately manipulate these pixel values since they have no prior knowledge about trigger location. Accordingly, we explore what will happen if we scale up all pixel values of benign and poisoned images.

**Settings.** In this section, we adopt BadNets (Gu et al., 2019)) and ISSBA (Li et al., 2021d) as the example for our discussion. They are representative of patch-based and non-patch-based backdoor attacks, respectively. We conduct experiments on the CIFAR-10 dataset (Krizhevsky, 2009) with ResNet (He et al., 2016). For both attacks, we inject a large number of poisoned samples to ensure a high attack success rate ($\geq 99\%$). For each benign and poisoned image, we gradually enlarge its pixel values with multiplication. We calculate the *average confidence* defined as the average probabilities of samples on the originally predicted label. In particular, we select the label predicted upon the original sample as the originally predicted label for each varied sample and constrain all pixel values within $[0, 1]$ during the multiplication process. More details are in our appendix.

**Results.** As shown in Figure 2, the average confidence scores of both benign and poisoned samples decrease during the multiplication process under the benign model. In other words, the predictions of modified benign and poisoned samples are changed during this process. In contrast, poisoned and benign samples have significantly distinctive behaviors under attacked models. Specifically, the average confidence of benign samples decreases whereas that of poisoned samples is relatively stable with the increase of multiplication times. We call this phenomenon as *scaled prediction consistency*.

To further explain this intriguing phenomenon (*i.e.*, scaled prediction consistency), we exploit recent studies on neural tangent kernel (NTK) (Jacot et al., 2018) for analyzing the backdoor-infected models inspired by (Guo et al., 2022c), as follows:

**Theorem 1.** *Suppose the poisoned training dataset consists of $N_b$ benign samples and $N_p$ poisoned samples, i.i.d. sampled from uniform distribution and belonging to $K$ classes. Assume that deep neural network $f(\cdot; \theta)$ be a multivariate kernel regression (RBF kernel) with the objective in Eq. (1). For a given attacked sample $\boldsymbol{x}' = (\boldsymbol{1} - \boldsymbol{m}) \odot \boldsymbol{x} + \boldsymbol{m} \odot \boldsymbol{t}$, we have: $\lim_{N_p \to N_b} C(n \cdot \boldsymbol{x}') = y_t, n \geq 1$.*

In general, Theorem 1 reveals that when the amount of poisoned samples closes to the benign samples or the attacked DNN over-fits the poisoned samples, it will still constantly predict the scaled attacked samples (*i.e.*, $n \cdot \boldsymbol{x}'$) as the target label $y_t$. Its proof can be found in Appendix A.

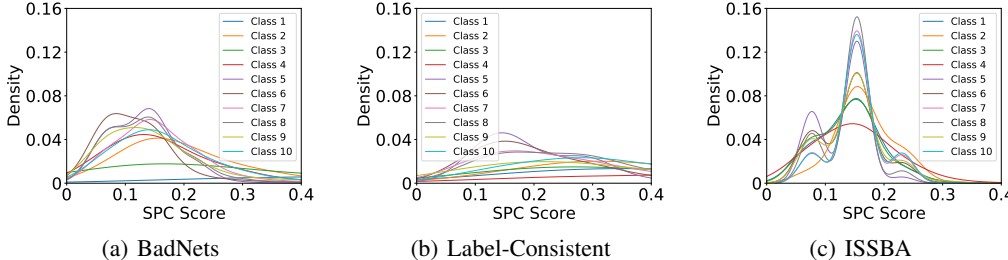

Figure 3: The SPC scores of benign samples from different classes under attacked models.

# 4    SCALED PREDICTION CONSISTENCY ANALYSIS (SCALE-UP)

Motivated by the phenomenon of scaled prediction consistency demonstrated in the previous section, we propose a simple yet effective black-box input-level backdoor detection, dubbed scaled prediction consistency analysis (SCALE-UP), in this section.

## 4.1    PRELIMINARIES

**Defender's Goals.** In general, defenders have two main goals, including *effectiveness* and *efficiency*. Effectiveness requires that the detection method can accurately identify whether a given image is malicious or not. Efficiency ensures that detection time is limited and therefore the deployed model can provide final results on time after the detection and prediction process.

**Threat Model.** We consider backdoor detection under the black-box setting in machine learning as a service (MLaaS) applications. Specifically, defenders can only query the third-party deployed model and obtain its predictions. They do not have any prior information about the backdoor attack or the model. In particular, we consider two data settings, including **1)** *data-free detection* and **2)** *data-limited detection*. The former one assumes that defenders have no holding benign samples, while the latter one allows defenders to have a few benign samples from each class. Note that we only assume to have the predicted label instead of the predicted probability vector in our method.

## 4.2    DATA-FREE SCALED PREDICTION CONSISTENCY ANALYSIS

As demonstrated in Section 3, we can use the average probability on the originally predicted label across its scaled images to determine whether a given suspicious image is malicious. In general, the larger the probability, the more likely the sample is poisoned. However, we can only obtain predicted labels while predicted probability vectors are inaccessible under our settings. Accordingly, we propose to examine whether the predictions of scaled samples are consistent, as follows:

Let $\mathcal{S}$ denotes a defender-specified scaling set (*e.g.*, $\mathcal{S} = \{3, 5, 7, 9, 11\}$). For a given input image $\boldsymbol{x}$ and the deployed classifier $C$, we define its *scaled prediction consistency (SPC)* as the proportion of labels of scaled images that are consistent with that of the input image, *i.e.*,

$$SPC(\boldsymbol{x}) = \frac{\sum_{n \in \mathcal{S}} \mathbb{I}\{C(n \cdot \boldsymbol{x}) = C(\boldsymbol{x})\}}{|\mathcal{S}|}, \quad (2)$$

where $\mathbb{I}$ is the indicator function and $|\mathcal{S}|$ denotes the size of scaling set $\mathcal{S}$. In particular, we constrain $n \cdot \boldsymbol{x} \in [0, 1]$ during the multiplication process.

Once we obtain the SPC value of suspicious input $\boldsymbol{x}$, we can determine it is malicious based on defender-specified threshold $T$. If $SPC(\boldsymbol{x}) > T$, we deem it as a backdoor sample.

## 4.3    DATA-LIMITED SCALED PREDICTION CONSISTENCY ANALYSIS

In our data-free scaled prediction consistency analysis, we treat all labels equally. However, we notice that the SPC values of benign samples under attacked models are different across classes (as shown in Figure 3). In other words, some classes are more consistent against image scaling compared to the remaining ones. These benign samples with have high SPC values may be mistakenly treated as malicious samples, leading to relatively low precision of our method.

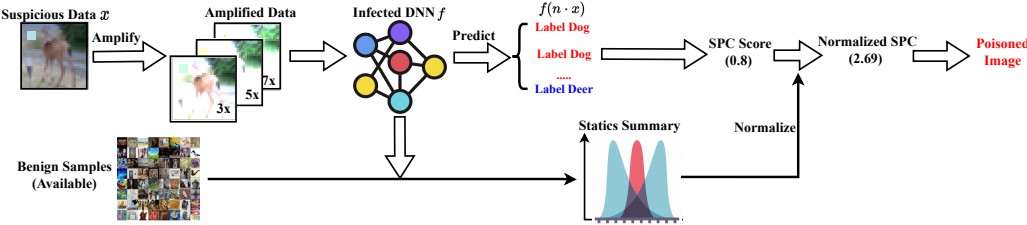

Figure 4: The main pipeline of our (data-limited) SCALE-UP. For each suspicious input, it first generates a set of its amplified images. After that, it takes amplified images to query the deployed DNN and obtain their predicted labels. Thirdly, we compute and normalize the SPC value based on the results and that of some local benign samples. SCALE-UP treats the input as a malicious image if the normalized SPC value is greater than a defender-specified threshold $T$.

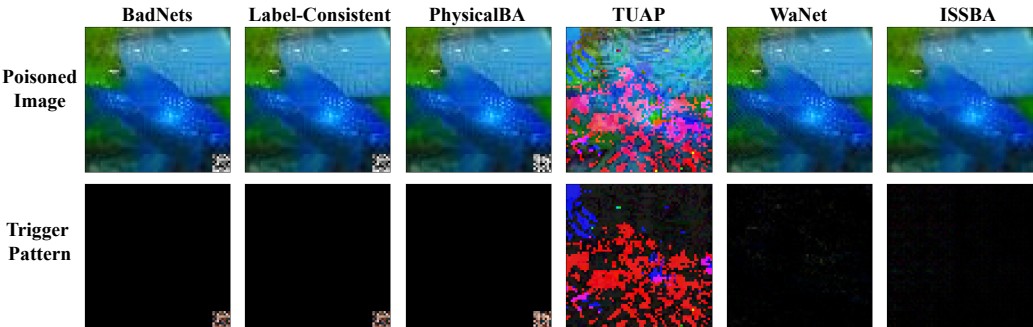

Figure 5: The demonstration of various trigger patterns of attacks used in our experiments.

In data-limited scaled prediction consistency analysis, we assume that defenders have a few benign samples from each class. This setting has been widely used in existing backdoor defenses (Li et al., 2021b; Guo et al., 2022c; Zeng et al., 2022). In this paper, we propose to exploit a set of statics summary (*i.e.*, mean $\mu$ and standard deviation $\sigma$) of these local benign samples to alleviate this problem, inspired by (Ioffe & Szegedy, 2015). We first estimate the statics summary for SPC values on samples from different classes, based on which to normalize the SPC value of suspicious input images according to their predicted labels. Specifically, for each class $i$, we calculate its corresponding mean $\mu_i$ and standard deviation $\sigma_i$ based on samples $\boldsymbol{X}_i$ belonging to class $i$, as follows:

$$\mu_i = \mathbb{E}_{\boldsymbol{x} \in \boldsymbol{X}_i}[SPC(\boldsymbol{x})], \quad \sigma_i = \sqrt{\mathbb{E}_{\boldsymbol{x} \in \boldsymbol{X}_i}[(\boldsymbol{x} - \mu_i)^2]}. \tag{3}$$

In the detection process, given a suspicious image $\boldsymbol{x}$ and the deployed model $C$, we normalize the SPC value generated by data-free SCALE-UP based on its predicted label $\hat{y} \triangleq C(\boldsymbol{x})$, as follows:

$$NSPC(\boldsymbol{x}) \triangleq SPC(\boldsymbol{x}) - \frac{\mu_{\hat{y}}}{\sigma_{\hat{y}}}. \tag{4}$$

Besides, for a more stable and effective performance, we automatically balance two terms in Eq. (4) to make their values at the same level. The main pipeline of our method is summarized in Figure 4.

## 5 EXPERIMENTS

### 5.1 MAIN SETTINGS

**Dataset and DNN Selection.** Following the settings in existing backdoor defenses, we conduct experiments on CIFAR-10 (Krizhevsky, 2009) and (Tiny) ImageNet (Russakovsky et al., 2015) datasets with ResNet (He et al., 2016). Please find more detailed information in our appendix.

**Attack Baselines.** In this paper, we evaluate our methods under six representative attacks, including **1)** BadNets (Gu et al., 2019), **2)** label consistent backdoor attack (dubbed 'Label-Consistent')

Table 1: The performance (AUROC) on the CIFAR-10 dataset. Among all different methods, the best result is marked in boldface while the value with underline denotes the second-best result. The failed cases (*i.e.*, AUROC $<0.55$) are marked in red. Note that STRIP require obtaining predicted probability vectors while other methods only need the predicted labels.

| Attack→
Defense↓ | BadNets | Label-Consistent | PhysicalBA | TUAP | WaNet | ISSBA | **Average** |
|---|---|---|---|---|---|---|---|
| STRIP | **0.989** | 0.941 | **0.971** | 0.671 | 0.475 | 0.498 | 0.758 |
| ShrinkPad | 0.951 | **0.957** | 0.631 | **0.869** | 0.531 | 0.513 | 0.742 |
| DeepSweep | 0.967 | 0.921 | 0.946 | 0.743 | 0.506 | 0.729 | 0.802 |
| Frequency | 0.891 | 0.889 | 0.881 | 0.851 | 0.461 | 0.497 | 0.745 |
| Ours (data-free) | 0.971 | 0.947 | 0.969 | 0.816 | 0.918 | **0.945** | 0.928 |
| Ours (data-limited) | 0.971 | 0.954 | 0.970 | 0.830 | **0.925** | **0.945** | **0.933** |

Table 2: The performance (AUROC) on the Tiny ImageNet dataset. Among all different methods, the best result is marked in boldface while the value with underline denotes the second-best result. The failed cases (*i.e.*, AUROC $<0.55$) are marked in red. Note that STRIP require obtaining predicted probability vectors while other methods only need the predicted labels.

| Attack→
Defense↓ | BadNets | Label-Consistent | PhysicalBA | TUAP | WaNet | ISSBA | **Average** |
|---|---|---|---|---|---|---|---|
| STRIP | **0.959** | **0.939** | **0.959** | 0.638 | 0.501 | 0.471 | 0.745 |
| ShrinkPad | 0.871 | 0.938 | 0.672 | **0.866** | 0.498 | 0.492 | 0.737 |
| DeepSweep | 0.951 | 0.930 | 0.939 | 0.759 | 0.503 | 0.714 | 0.799 |
| Frequency | 0.864 | 0.859 | 0.864 | 0.837 | 0.428 | 0.540 | 0.732 |
| Ours (data-free) | 0.936 | 0.904 | 0.939 | 0.763 | 0.943 | 0.948 | 0.905 |
| Ours (data-limited) | 0.947 | 0.911 | 0.939 | 0.763 | **0.946** | **0.949** | **0.909** |

(Turner et al., 2019), **3)** physical backdoor attack (dubbed 'PhysicalBA') (Li et al., 2021c), **4)** clean-label backdoor attack with targeted universal adversarial perturbation (dubbed 'TUAP') (Zhao et al., 2020a), **5)** WaNet (Nguyen & Tran, 2021), and **6)** ISSBA (Li et al., 2021d). They are the representative of patch-based and non-patch-based backdoor attacks under different settings. For each attack, we randomly select the target label and inject sufficient poisoned samples to ensure the attack success rate $\geq 98\%$ while preserving the overall model performance. We implement these attacks based on the open-sourced backdoor toolbox (Li et al., 2023). We demonstrate the trigger patterns of adopted attacks for Tiny ImageNet in Figure 5. More detailed settings are in the appendix.

**Defense Baselines.** In this paper, we focus on the backdoor detection under the black-box setting where defenders can only query the deployed model and obtain its predicted label. Accordingly, we compare our methods to ShrinkPad (Li et al., 2021c), DeepSweep (Qiu et al., 2021), and artifacts detection in the frequency domain (dubbed 'Frequency') (Zeng et al., 2021). We also compare our methods to STRIP (Gao et al., 2021) that requires additional requirement (*i.e.*, obtaining predict probability vectors). We assume that defenders have 100 benign samples per class under the data-limited setting. Please find more defense details in our appendix.

**Settings for Evaluation Datasets.** Following the previous work (Lee et al., 2018), we use a positive (*i.e.*, attacked) and a negative (*i.e.*, benign) dataset to evaluate each defense. Specifically, the positive dataset contains the attacked testset and its augmented version, while the negative dataset contains a benign testset and its augmented version. The augmented datasets are created by adding small random noise to their original version. The noise magnitude is set to 0.05. In particular, adding these random noises will not significantly affect the attack success rate and the benign accuracy of deployed models. The introduction of the augmented datasets is to prevent evaluated defenses from over-fitting the benign or the poisoned testsets.

**Evaluation Metrics.** Following existing detection-based backdoor defenses (Gao et al., 2021; Guo et al., 2022c), we adopt the area under receiver operating curve (AUROC) (Fawcett, 2006) to evaluate defense effectiveness, while use the inference time for evaluating efficiency. In general, the higher the AUROC and the lower the inference time, the better the backdoor detection.

## 5.2 Main Results

As shown in Table 1-2, *all baseline detection methods fail in defending against some evaluated attacks*. Specifically, they have relatively low AUROC in detecting advanced non-patch-based attacks (*i.e.*, WaNet and ISSBA). This failure is mostly because these defenses have some implicit assumptions (*e.g.*, the trigger pattern is sample-agnostic or static) about the attack, which are not necessarily true in practice. In contrast, *our methods reach promising performance in all cases on both datasets*. For example, the AUROC of our data-limited SCALE-UP is 0.5 greater than all baseline defenses in detecting WaNet on the Tiny ImageNet dataset. Even under the classical patch-based attacks (*i.e.*, BadNets, Label-Consistent, and PhysicalBA), the effectiveness of our methods is on par with or bet-

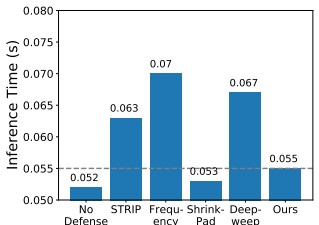

Figure 6: The inference time on the CIFAR-10 dataset.

ter than all baseline defenses. Our methods are even better than STRIP, which requires obtaining predicted probability vectors instead of predicted labels. We also provide the ROC curves of defenses against all attacks in Appendix N. These results verify the effectiveness of our defenses.

Besides, we also calculate the inference time of all defenses under the same computational facilities. In particular, we calculate the inference time of methods requiring to obtain the predictions of multiple images by feeding them simultaneously (in a data batch) into the deployed model instead of predicting them one by one. Besides, we only report the inference time of our SCALE-UP under the data-limited setting, since both of them have very similar running times. As shown in Figure 6, our method requires fewer inference times compared to almost all baseline defenses. The only exception is ShrinkPad, whereas its effectiveness is significantly lower than that of our method. Our detection is approximately 5% slower compared with the standard inference process without any defense. These results show the efficiency of our SCALE-UP detection.

## 5.3 Discussion

In this section, we discuss whether our method is still effective under different (adversarial) settings.

### 5.3.1 Defending against Attacks with Larger Trigger Sizes

Recent studies (Qiu et al., 2021) revealed that some defenses (*e.g.*, ShrinkPad) may fail in detecting samples with a relatively large trigger size. In this part, we use two patch-based attacks (*e.g.*, BadNets and PhysicalBA) on the Tiny ImageNet dataset for discussion. As shown in Figure 7(a), our methods have high AUROC values ($> 0.93$) across different trigger sizes under both data-free and data-limited settings, although there are some mild fluctuations. These results verify the resistance of our SCALE-UP detection to adaptive attacks with large trigger patterns.

### 5.3.2 The Resistance to Potential Adaptive Attacks

Most recently, (Qi et al., 2023) demonstrated that reducing the poisoning rate is a simple yet effective method to design adaptive attacks for detection-based defenses, since it can reduce the differences between benign and poisoned samples. Motivated by this finding, we first explore whether our SCALE-UP methods are still effective in defending against attacks with low poisoning rates. We adopt BadNets on the CIFAR-10 dataset as an example for our discussions. In particular, we report the results of all poisoned testing samples and those that can be predicted as the target label, respectively. We design this setting since attacked models may still correctly predict many poisoned samples even if they contain trigger patterns when the poisoning rate is relatively low.

As shown in Figure 7(b), the attack success rate (ASR) increases with the increase of the poisoning rate. Our method can still correctly detect poisoned samples that can successfully attack the deployed model even when the poisoning rate is set to $0.4\%$ where the ASR is lower than 70%. In these cases, the AUROC $> 0.95$. Besides, our method can still reach promising performance (AUROC $> 0.8$) in detecting all poisoned samples. These results verify the resistance of our defense to adaptive attacks with low poisoning rates, where attacked models don't over-fit backdoor triggers.

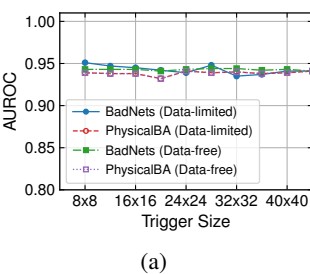 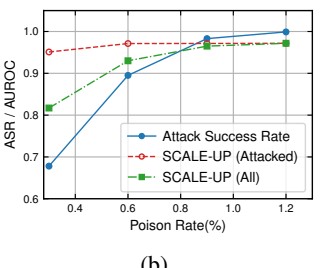 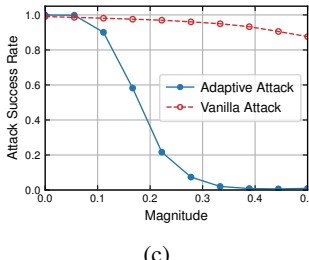

(a)                           (b)                           (c)

Figure 7: The results of additional experiments in our discussion. **(a)** The performance of our methods under attacks with different trigger sizes. **(b)** The attack performance and the defense effectiveness on all poisoned testing samples and those that can successfully attack the deployed model. **(c)** The effectiveness of adaptive and vanilla backdoor attacks on poisoned samples with random noise under different magnitudes.

To further evaluate the resistance of our SCALE-UP to potential adaptive methods, we evaluate it under the worst scenario, where the backdoor adversaries are fully aware of our mechanism. Specifically, we design a strong adaptive attack by introducing an additional defense-resistant regularization term to the vanilla attack illustrated in Eq. (1). This regularization term is used to prevent scaled poisoned samples $n \cdot \boldsymbol{x}'_j$ being predicted as the target label $y_t$, as follows:

$$\min_{\boldsymbol{\theta}} \sum_{i=1}^{N_b} \mathcal{L}(f(\boldsymbol{x}_i; \boldsymbol{\theta}), y_i) + \sum_{j=1}^{N_p} \mathcal{L}(f(\boldsymbol{x}'_j; \boldsymbol{\theta}), y_t) + \sum_{j=1}^{N_p} \mathcal{L}(f(n \cdot \boldsymbol{x}'_j; \boldsymbol{\theta}), y_j). \tag{5}$$

Similar to previous experiments, we adopt BadNets to design the adaptive attack on the CIFAR-10 dataset. As we expected, this method can bypass our detection resulting in a low AUROC (*i.e.*, 0.467). However, *the adaptive attack would make the poisoned samples significantly more vulnerable to small random Gaussian noises*. As shown in Figure 7(c), random noises with a small magnitude ($< 0.3$) will significantly reduce the attack success rate of the adaptive attack, while having minor adverse effects on the vanilla attack. In other words, defenders can easily adopt random noises to defend against this adaptive attack. We speculate that its vulnerability is mostly because the regularization term significantly constrains the generalization of attacked DNNs on the poisoned samples. We will further explore its intrinsic mechanism in our future work.

### 5.3.3 THE EFFECTIVENESS OF SCALING PROCESS

Technically, the scaling process in our SCALE-UP detection can be regarded as a data augmentation method generating different modified versions of the suspicious input image. It naturally raises an intriguing question: *If other augmentation methods are adopted, is our SCALE-UP detection still effective?* Since flip operations and frequency domain analysis have been adopted in (Li et al., 2021c; Zeng et al., 2021) for defense and proved to have minor benefits to detecting advanced backdoor attacks (Li et al., 2021d; Nguyen & Tran, 2021), we investigate the effectiveness of adding increasing magnitudes of random noise. Due to the limitations of space, we include the detailed experimental design and evaluation in Appendix O.

## 6 CONCLUSION

In this paper, we proposed a simple yet effective black-box input-level backdoor detection (dubbed SCALE-UP) that can be used in real-world applications under the machine learning as a service (MLaaS) setting. Our method was motivated by an intriguing new phenomenon (dubbed *scaled prediction consistency*) that the predictions of poisoned samples are significantly more consistent compared to those of benign ones when amplifying all pixel values. We also provided theoretical foundations to explain this phenomenon. In particular, we designed our SCALE-UP detection method under both data-free and data-limited settings. Extensive experiments on benchmark datasets verified the effectiveness and efficiency of our method and its resistance to potential adaptive attacks.

## ETHICS STATEMENT

DNNs have been widely and successfully adopted in many mission-critical applications. Accordingly, their security is of great significance. The existence of backdoor threats raises serious concerns about using third-party models under the machine learning as a service (MLaaS) setting. In this paper, we propose a simple yet effective black-box input-level backdoor detection. Accordingly, this work has no ethical issues since it does not reveal any new security risks and is purely defensive. However, we need to notice that our methods can only be used to filter poisoned testing samples whereas they do not reduce the intrinsic backdoor vulnerability of deployed models. Our defense also couldn't recover trigger patterns. People should not be too optimistic about eliminating backdoor threats. We will further improve our method by exploring how to recover triggers.

## ACKNOWLEDGMENT

This work is mainly supported by Samsung Research America, Mountain View, CA and partially supported by NSF CNS 2135625, CPS 2038727, CNS Career 1750263, and a Darpa Shell grant.

## REPRODUCIBILITY STATEMENT

We have provided detailed information on datasets, models, training settings, and computational facilities in our main manuscript and appendix. The codes for reproducing our main experiments are also open-sourced at https://github.com/JunfengGo/SCALE-UP.

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

APPENDIX

## A  THE OMITTED PROOF OF THEOREM 1

**Theorem 1.** *Suppose the poisoned training dataset consists of $N_b$ benign samples and $N_p$ poisoned samples, i.i.d. sampled from uniform distribution and belonging to $K$ classes. Assume that deep neural network $f$ adopt RBF kernel and cross-entropy loss with the optimization objective in Eq.1. For a given attacked sample $\boldsymbol{x}' = (\mathbf{1} - \boldsymbol{m}) \odot \boldsymbol{x} + \boldsymbol{m} \odot \boldsymbol{t}$, we have:* $\lim_{N_p \to N_b} C(n \cdot \boldsymbol{x}') = y_t, n \geq 1$.

**Proof of Theorem 1:**  Following (Guo et al., 2022c), we have the regression solution for NTK is:

$$\phi_t(\cdot) = \frac{\sum_{i=1}^{N_b} K(\cdot, \boldsymbol{x_i}) \cdot y_i + \sum_{i=1}^{N_p} K(\cdot, \boldsymbol{x_i'}) \cdot y_t}{\sum_{i=1}^{N_b} K(\cdot, \boldsymbol{x_i}) + \sum_{i=1}^{N_p} K(\cdot, \boldsymbol{x_i'})}, \tag{6}$$

where $\phi_t(\cdot) \in \mathbb{R}$ is the predictive probability output of $f(\cdot; \theta)$ for the target class $t$ and $y_i$ is the corresponding one-hot label. $K(\boldsymbol{x}, \boldsymbol{x_i}) = e^{-2\gamma||\boldsymbol{x}-\boldsymbol{x_i}||^2}$ ($\gamma > 0$). Since the training samples are evenly distributed, there are $\frac{N_b}{k}$ benign samples belonging to $y_t$. Without loss of generality, we assume the target label $y_t = 1$ while others are 0. Then the regression solution can be converted to:

$$\phi_t(\cdot) = \frac{\sum_{i=1}^{N_b/k} K(\cdot, \boldsymbol{x_i}) + \sum_{i=1}^{N_p} K(\cdot, \boldsymbol{x_i'})}{\sum_{i=1}^{N_b} K(\cdot, \boldsymbol{x_i}) + \sum_{i=1}^{N_p} K(\cdot, \boldsymbol{x_i'})}, \tag{7}$$

For a given backdoored sample $\boldsymbol{x}' = (1 - m) \odot \boldsymbol{x} + m \odot \boldsymbol{t}$, we can simplify Eq. (7) as:

$$\phi_t(\boldsymbol{x}') \geq \frac{\sum_{i=1}^{N_p} K(\boldsymbol{x}', \boldsymbol{x_i'})}{\sum_{i=1}^{N_b} K(\boldsymbol{x}', \boldsymbol{x_i}) + \sum_{i=1}^{N_p} K(\boldsymbol{x}', \boldsymbol{x_i'})}, \tag{8}$$

we here remove the term $\sum_{i=1}^{N_b/k} K(\boldsymbol{x}', \boldsymbol{x_i})$. This is because $\boldsymbol{x}'$ typically don't belong to the target $y_t$ and $\sum_{i=1}^{N_b/k} K(\boldsymbol{x}', \boldsymbol{x_i}) << \sum_{i=1}^{N_p} K(\boldsymbol{x}', \boldsymbol{x_i'})$, otherwise the attacker has no incentive to craft poisoned sample.

When $N_p$ close to $N_b$, which implies that the poisoning rate close to $50\%$, the attacker can achieve the optimal attack efficacy (Liu et al., 2018b; Gu et al., 2019; Li et al., 2021d). Given $K(\boldsymbol{x}, \boldsymbol{x_i}) = e^{-2\gamma||\boldsymbol{x}-\boldsymbol{x_i}||^2}$ ($\gamma > 0$), if $N_p = N_b$, we have:

$$\phi_t(n \cdot \boldsymbol{x}') = \frac{\sum_{i=1}^{N_p} K(n \cdot \boldsymbol{x}', \boldsymbol{x_i'})}{\sum_{i=1}^{N_b} K(n \cdot \boldsymbol{x}', \boldsymbol{x_i}) + \sum_{i=1}^{N_p} K(n \cdot \boldsymbol{x}', \boldsymbol{x_i'})}. \tag{9}$$

If $n = 1$, we can easily obtain that:

$$\sum_{i=1}^{N_p} e^{-2\gamma||(1-m)\odot(\boldsymbol{x}-\boldsymbol{x_i})||^2} - e^{-2\gamma||(1-m)\odot(\boldsymbol{x}-\boldsymbol{x_i})+m\odot(\boldsymbol{t}-\boldsymbol{x_i})||^2} \tag{10}$$

$$= \sum_{i=1}^{N_p} e^{-2\gamma||(1-m)\odot(\boldsymbol{x}-\boldsymbol{x_i})||^2} (1 - e^{-2\gamma||m\odot(\boldsymbol{t}-\boldsymbol{x_i})||^2}) > 0. \tag{11}$$

Since the internal term $(1 - e^{-2\gamma||m\odot(\boldsymbol{t}-\boldsymbol{x_i})||^2})$ can be always larger than 0, thus it is clear that $f(\boldsymbol{x}') = y_t$, which is also consistent with the practice.

However, when $n > 1$, to compare $\sum_{i=1}^{N_p} K(n \cdot \boldsymbol{x}', \boldsymbol{x}_i')$ and $\sum_{i=1}^{N_b} K(n \cdot \boldsymbol{x}', \boldsymbol{x}_i)$, we have:

$$\sum_{i=1}^{N_o} K(n \cdot \boldsymbol{x}', \boldsymbol{x}_i') - \sum_{i=1}^{N_b} K(n \cdot \boldsymbol{x}', \boldsymbol{x}_i) \tag{12}$$

$$= \sum_{i=1}^{N_b} e^{-2\gamma||(1-m)\odot(n\cdot\boldsymbol{x}-\boldsymbol{x}_i)+m\odot(n-1)\boldsymbol{t}||^2} - e^{-2\gamma||(1-m)\odot(n\cdot\boldsymbol{x}-\boldsymbol{x}_i)+m\odot(n\cdot\boldsymbol{t}-\boldsymbol{x}_i)||^2} \tag{13}$$

$$= \sum_{i=1}^{N_b} e^{-2\gamma||(1-m)\odot(n\cdot\boldsymbol{x}-\boldsymbol{x}_i)+m\odot(n-1)\boldsymbol{t}||^2} (1 - e^{-2\gamma(||m\odot(n\cdot\boldsymbol{t}-\boldsymbol{x}_i)||^2 - ||m\odot(n-1)\boldsymbol{t}||^2)}). \tag{14}$$

$$\tag{15}$$

Regarding the internal term $||m \odot (n \cdot \boldsymbol{t} - \boldsymbol{x}_i)||^2 - ||m \odot (n-1)\boldsymbol{t})||^2$, we have:

$$||m \odot (n \cdot \boldsymbol{t} - \boldsymbol{x}_i)||^2 - ||m \odot (n-1)\boldsymbol{t})||^2 \tag{16}$$

$$= \sum_{j,k \in \ trigger} ||(n-1) \cdot \boldsymbol{t}_{j,k} + (\boldsymbol{t}_{j,k} - \boldsymbol{x}_{i,j,k})||^2 - ||(n-1) \cdot \boldsymbol{t}_{j,k}||^2 \tag{17}$$

$$= \sum_{j,k \in \ trigger} \delta_{i,j,k}^2 + 2(n-1) \cdot \delta_{i,j,k} \boldsymbol{t}_{j,k}, \tag{18}$$

where $\delta_{i,j,k}$ is the pixel-level residue between the trigger and benign samples. We assume that the $\delta_{i,j,k} \boldsymbol{t}_{j,k}$ close to a zero mean for inputs, thus we can rewrite Eq.(12) as follows:

$$\sum_{i=1}^{N_p} K(n \cdot \boldsymbol{x}', \boldsymbol{x}_i') - \sum_{i=1}^{N_b} K(n \cdot \boldsymbol{x}', \boldsymbol{x}_i) \tag{19}$$

$$\approx \sum_{i=1}^{N_p} e^{-2\gamma||(1-m)\odot(n\cdot\boldsymbol{x}-\boldsymbol{x}_i)+m\odot(n-1)\boldsymbol{t}||^2} (1 - e^{-2\gamma \sum_{j,k \in trigger} \delta_{i,j,k}^2}) \tag{20}$$

$$> 0. \tag{21}$$

Put Eq. (19) and Eq. (9) together, we know that $\phi_t(n \cdot \boldsymbol{x}') \geq 0.5$, as $N_p \to N_b$, we have:

$$\lim_{N_p \to N_b} C(n \cdot \boldsymbol{x}_t') = y_t, n \geq 1.$$

$\square$

## B    THE DETAILED CONFIGURATIONS OF THE EMPIRICAL STUDY

We adopt BadNets (Gu et al., 2019)) and ISSBA (Li et al., 2021d) as the example for our discussion. They are the representative of patch-based and non-patch-based backdoor attacks, respectively. We conduct experiments on the CIFAR-10 dataset (Krizhevsky, 2009) with ResNet-34 (He et al., 2016). For both attacks, we inject a large number of poisoned samples to ensure a high attack success rate ($\geq 99\%$). For each benign and poisoned image, we gradually enlarge its pixel values with multiplication. We calculate the *averaged confidence* defined as the averaged probabilities of samples on the originally predicted label. In particular, we select the label predicted upon the original sample as the originally predicted label for each varied sample and constrain all pixel values within $[0, 1]$ during the multiplication process. In particular, we follow previous works (Gu et al., 2019; Li et al., 2021d) to implement the backdoor attacks. Specifically, the trigger for BadNets is a $4 \times 4$ square consisting of random pixel values; the trigger of ISSBA is generated via DNN-based image steganography (Tancik et al., 2020). Both attacks are implemented via `BackdoorBox` (Li et al., 2023).

Table 3: Detailed information about the adopted datasets.

| Dataset | # Classes | Image Size | # Training Images |
|---------|-----------|------------|-------------------|
| CIFAR-10 | 10 | $3 \times 32 \times 32$ | 50,000 |
| Tiny ImageNet | 200 | $3 \times 64 \times 64$ | 1,000,000 |

## C   THE DETAILS FOR TRAINING ATTACKED MODELS

We train backdoor-infected models using BackdoorBox (Li et al., 2023). We set the training epoch as 200 and the poisoning rate as $5 - 10\%$ for each attack to ensure a high attack success rate. In particular, except for PhysicalBA, we don't involve additional data augmentation in training infected models as we want to better reveal the properties of various backdoor approaches. For each infected model, we randomly select infected labels to ensure their predictions on benign inputs are similar to the benign models, which ensures the stealthiness of backdoor attacks. Regarding the data-limited scenario and ablation study, we intentionally affect multiple labels to ensure the infected models similar to benign models except for the Trojan behaviors. Specifically, we inject less amount($\leq 5\%$) of poisoned samples to affect labels other than the target label. This is because previous work (Guo et al., 2022c) found that certain dense backdoor attacks (*e.g.*, ISSBA, WaNet) would make the infected DNNs sensitive to noisy or out-of-distribution samples on CIFAR-10 dataset. Accordingly,they are less stealthy and are easy to be detected during the sampling process of the data-limited setting and settings of our ablation study. As such, in these settings, to evaluate SCALE-UP in a rather practical scenario, we train infected DNNs to have similar behaviors on noisy samples as the benign DNNs. The details for each dataset are included in Table 3.

### C.1   THE ACCURACY AND ATTACK SUCCESS RATE (ASR) FOR EVALUATED MODELS

The accuracy and ASR for the evaluated models for each task in included in Table 4.

Table 4: The BA and ASR for the evaluated models on each dataset.

| Task↓ Model→ | Infected Model | | Normal Model Accuracy |
|--------------|------|------|----------------------|
| | BA | ASR | |
| CIFAR-10 | $\geq 90.04\%$ | $\geq 97.7\%$ | $\geq 92.31\%$ |
| Tiny ImageNet | $\geq 36.98\%$ | $\geq 97.22\%$ | $\geq 40.11\%$ |

Table 5: The performance of six defense baselines against partial backdoor attacks

| Task↓   Attack → | STRIP | ShrinkPad | Frequency | DeepSweep | Ours (data-free) | Ours (data-limited) |
|------------------|-------|-----------|-----------|-----------|------------------|---------------------|
| CIFAR-10 | 0.617 | 0.949 | 0.891 | 0.967 | 0.971 | 0.971 |
| Tiny ImageNet | 0.601 | 0.868 | 0.861 | 0.951 | 0.936 | 0.971 |

## D   THE DETAILED CONFIGURATIONS FOR BASELINE DEFENSES

- **STRIP:** We implement STRIP following their official open-sourced codes[*].
- **ShrinkPad:** We implement ShrinkPad following their official open-sourced codes[†].
- **Frequeny:** We implement Frequency approach following their official codes[‡].
- **DeepSweep:** We implement DeepSweep using `Scipy` package to remove the high-frequency noise and use `torchvision.transforms` and `keras.preprocess` packages to conduct transformation to inputs. Notably, we don't apply `finetune` process within DeepSweep since we only focus on the black-box detection scenarios.

---

[*] https://github.com/garrisongys/STRIP.git
[†] https://github.com/THUYimingLi/BackdoorBox.git
[‡] https://github.com/YiZeng623/frequency-backdoor.git

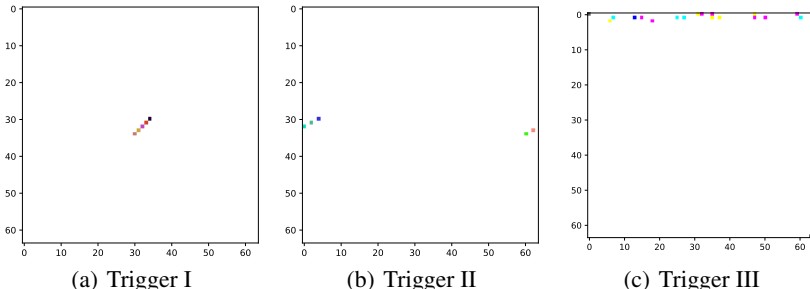

(a) Trigger I      (b) Trigger II      (c) Trigger III

Figure 8: The demonstration of dynamic triggers.

# E THE DESCRIPTIONS FOR MAIN EVALUATION METRIC

- The receiver operating curve (ROC) shows the trade-off between detection the success rate for poisoned samples and detection error rate for benign samples across different decision thresholds $T$ under infected-DNNs.
- Inference Time: we implement each approach under the platform with one NVIDIA GPU 1080 Ti and a Intel(R) Xeon(R) CPU E5-2650 v4 @ 2.20GHz with batch size $= 1$. We test the inference time of each approach with an average of 1,000 runs.

# F SETTINGS FOR MEASURING THE INFERENCE TIME

Since we focus on defending against backdoor attacks in the inference phase, we here measure the inference time by:

- Identifying whether the input sample is poisoned or not.
- If the input is a benign sample, we next should use the target model to predict it.

For STRIP, ShrinkPad, DeepSweep, and SCALE-UP, we leverage the target model's prediction on the (augmented) inputs for defense purpose, which means the input can be identified and predicted at the same time. As for Frequency, which leverages a secondary neural network to predict the frequency domain of each given input. However, if the input is identified as benign, the target DNNs should also deliver prediction on it. We here assume the benign and poisoned samples have equal possibilities. Therefore, we calculate the inference time for Frequency as follows:

$$\text{time} = \text{TIME}(\text{Frequency}(\text{input})) + 0.5 \cdot \text{TIME}(\text{DNN}(\text{input})). \tag{22}$$

While for other approaches, we measure their inference time via:

$$\text{time} = \text{TIME}(\text{DNN}((\text{Augumented}) \text{ INPUT}))). \tag{23}$$

In particular, we calculate the inference time of methods required to obtain the predictions of multiple images by feeding them simultaneously (in a batch) into the deployed model instead of predicting them one by one. This approach is feasible since defenders can easily and efficiently obtain all of them before feeding them into the deployed model.

# G PERFORMANCE UNDER MULTIPLE-BACKDOOR TRIGGERS WITHIN A SINGLE INFECTED LABEL

Consistent with (Guo et al., 2022c; Wang et al., 2019), we also evaluate the efficacy of SCALE-UP under a more challenging scenario where multiple backdoors are embedded within a single target label. We randomly select a label as the infected label and inject various types of poisoned samples in the training phase. We inject arbitrary amounts of poison samples for each backdoor trigger to ensure the attack efficacy $ASR \geq 99\%$. The demonstrations for used backdoor triggers are shown in Figure 8. Under such considered scenario, we evaluate our SCALE-UP on CIFAR-10 and Tiny ImageNet datasets using ResNet-34.

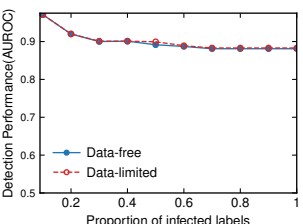

Figure 10: The performance of SCALE-UP for multiple infected labels.

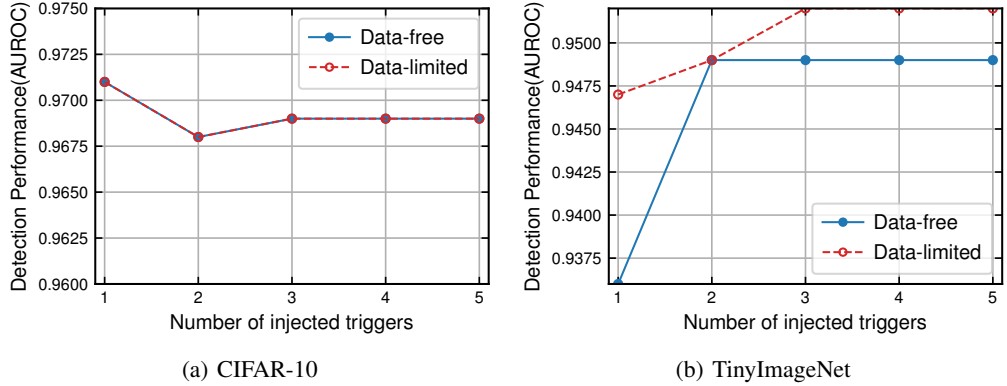

(a) CIFAR-10           (b) TinyImageNet

Figure 9: The average results for multiple triggers within a single label.

As shown in Figure 9, SCALE-UP performs resilient to the increasing number of injected backdoor triggers. This may be caused by infected models already generalized for backdoor triggers.

## H   THE PERFORMANCE AGAINST MULTIPLE INFECTED LABELS

We also test SCALE-UP under the scenario where the suspicious model has multiple infected labels. Under this scenario, we test SCALE-UP on CIFAR-10. This is because models on Tiny ImageNet would be prone to multiple infected labels, as reported by (Guo et al., 2022c), affecting more than $14\%$ labels can make the accuracy significantly drop$\geq 3\%$. We implement BadNets as backdoor attacks, the trigger size is $4 \times 4$. The results are shown in Figure 10. These results show that affecting multiple infected labels could slightly reduce the performance of SCALE-UP. Besides, the data-limited scenario performs better than the data-free scenario. However, even with $100\%$ labels are infected, SCALE-UP can still perform effectively with AUROC $\geq 0.883$.

## I   IMPACTS FOR THE NUMBER OF COEFFICIENTS

We test SCALE-UP on six attacks with varying $n$. We here use ResNet-34 on TinyImageNet for evaluation. As shown in Figure 11, we find that the performance of SCALE-UP increases along with $n$ increasing. Moreover, we find that SCALE-UP performs more sensitive on $n$ for the TUAP attack compared with other attack techniques. Moreover, we find that SCALE-UP performs similarly sensitive on $n$ in both data-limited and data-free settings. In most settings, with $n \geq 11$ SCALE-UP can achieve optimal performance on six different attacks.

## J   IMPACT FOR THE SIZE OF LOCAL SAMPLES

We also test the sensitivity of SCALE-UP on the size of local samples per label under the data-limited setting. We test SCALE-UP on Tiny ImageNet using ResNet-34 against six attacks. The results are shown in Figure 12. We can see that with the size of local samples increases, the performance of SCALE-UP improves and achieves optimal performance when the size $\geq 100$.

## K   PERFORMANCE UNDER SOURCE-LABEL-SPECIFIC BACKDOOR SCENARIOS

The Source-label-specific (Partial) backdoor scenarios is that the backdoor attacks can perform effectively when it applies to images of a certain specific class. Such a scenario makes backdoor attacks very hard to detect (Wang et al., 2019; Gao et al., 2021), thus the attacker may have a great incentive to implement such a backdoor attack in the real world. Therefore, we evaluate SCALE-UP under such a practical scenario and compare SCALE-UP with previous work. We test SCALE-UP using ResNet-34 on CIFAR-10 and Tiny ImageNet. As shown in Table 5, we find that most defense

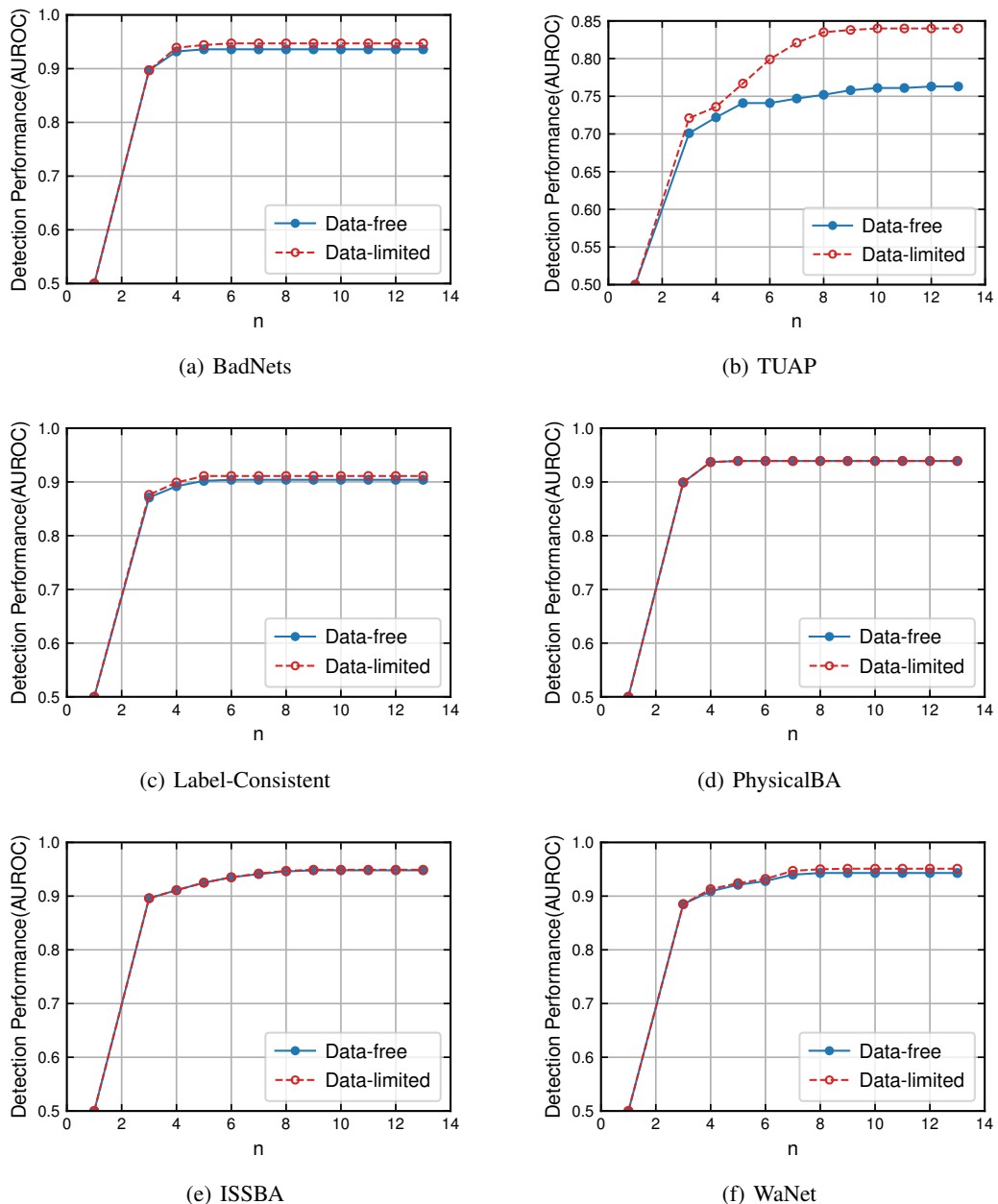

Figure 11: The impact for the coefficients $n$.

approaches perform resilient against the partial backdoor attack except STRIP (Gao et al., 2021). This is because STRIP assumes the trigger can perform effectively across various images. Under this scenario, SCALE-UP can outperform all baseline defenses.

## L  THE ROBUSTNESS OF SCALE-UP

Since SCALE-UP is an inference-phase backdoor defense approach, it is necessary to investigate the robustness of SCALE-UP on benign and poisoned samples. Following previous work (Du et al., 2020), we evaluate the robustness of our approach by testing different magnitudes of noisy inputs. Notably, we test SCALE-UP on benign and poisoned samples, respectively, which is because they exhibit different robustness under random noise as we show in Section 5.3.3. Moreover, the magnitudes of added random noise ensure the classification accuracy and attack success rate for benign and poisoned samples. The results are shown in Figure 13. We test our approach using ResNet-34

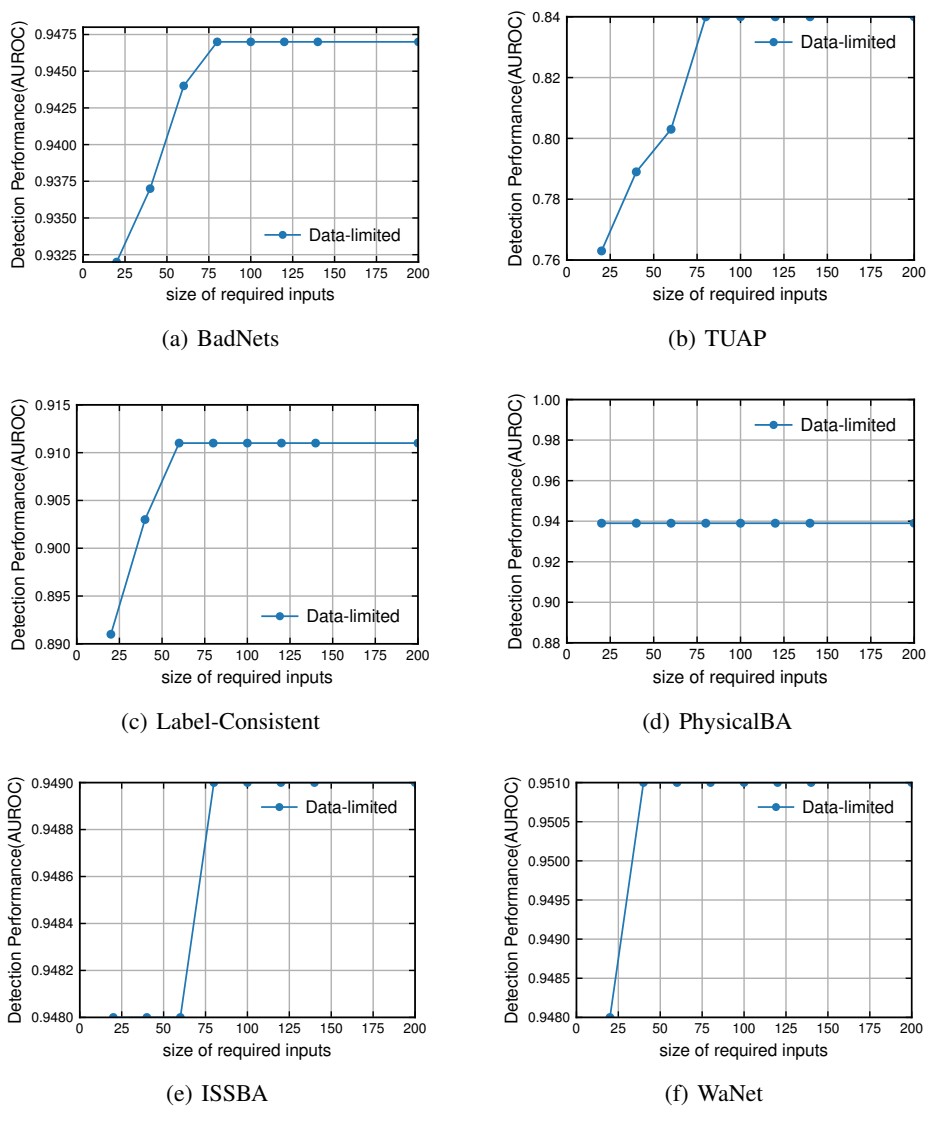

Figure 12: The impact for the size of required inputs.

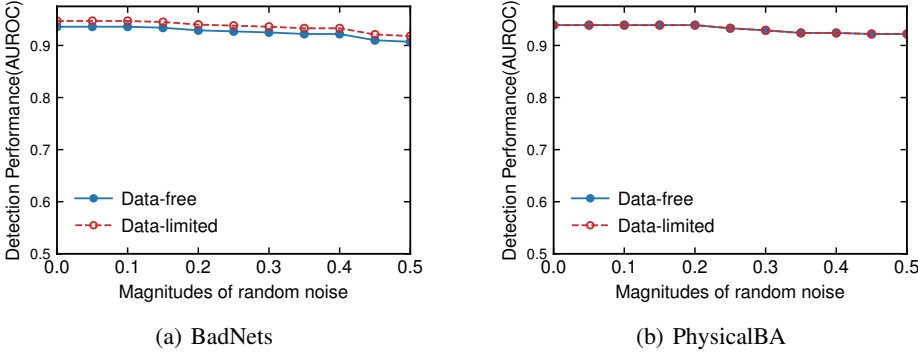

Figure 13: The robustness of SCALE-UP.

on the TinyImageNet task. The noise is randomly sampled from Gaussian distribution and we intentionally filter the failed poisoned samples. We only test BadNets and PhysicalBA since only these

Table 6: The performance (AUROC) on the Tiny ImageNet dataset under VGG-19. Among all different methods, the best result is marked in boldface while the value with underline denotes the second-best result. The failed cases (*i.e.*, AUROC $<$ 0.55) are marked in red. Note that STRIP requires obtaining predicted probability vectors while other methods only need the predicted labels.

| Attack→ Defense↓ | BadNets | Label-Consistent | PhysicalBA | TUAP | WaNet | ISSBA | Average |
|---|---|---|---|---|---|---|---|
| STRIP | **0.941** | 0.908 | **0.941** | 0.576 | 0.521 | 0.489 | 0.729 |
| ShrinkPad | 0.857 | **0.919** | 0.631 | 0.831 | 0.499 | 0.490 | 0.705 |
| DeepSweep | 0.939 | 0.907 | 0.921 | 0.744 | 0.511 | 0.711 | 0.788 |
| Frequency | 0.864 | 0.859 | 0.864 | 0.827 | 0.428 | 0.540 | 0.730 |
| Ours (data-free) | 0.936 | 0.846 | 0.907 | 0.858 | 0.893 | 0.767 | 0.868 |
| Ours (data-limited) | 0.936 | 0.851 | 0.907 | **0.888** | **0.904** | **0.836** | **0.887** |

Table 7: The performance (AUROC) of SCALE-UP variants with random noises on CIFAR-10 and Tiny-ImageNet datasets. The failed cases (*i.e.*, AUROC $<$ 0.55) are marked in red.

| Dataset↓ | Attack→ Setting↓ | BadNets | Label Consistent | PhysicalBA | TUAP | WaNet | ISSBA | Average |
|---|---|---|---|---|---|---|---|---|
| CIFAR-10 | data-free | 0.939 | 0.816 | 0.976 | 0.698 | 0.497 | 0.421 | 0.724 |
| | data-limited | 0.939 | 0.873 | 0.981 | 0.706 | 0.432 | 0.444 | 0.729 |
| Tiny ImageNet | data-free | 0.951 | 0.711 | 0.899 | 0.632 | 0.531 | 0.467 | 0.706 |
| | data-limited | 0.951 | 0.761 | 0.899 | 0.644 | 0.534 | 0.501 | 0.706 |

two attacks perform robustness against random noise, as illustrated in Section 5.3.3. We find that our approach is robust against noisy poisoned samples.

# M  ADDITIONAL RESULTS UNDER VGG ARCHITECTURE

In our main manuscript, we evaluate our method under the ResNet architecture. In this section, we conduct additional experiments under VGG-19 (BN) on Tiny ImageNet, to verify that the phenomenon of *scaled prediction consistency* is valid across different model architectures.

As shown in Figure 14, the scaled prediction consistency still holds in all cases. Specifically, the average confidence of benign samples decreases significantly faster than that of poisoned ones with the increase in multiplication time. Besides, as shown in Table 6, our methods are still better than all baseline defenses. These results verify the effectiveness of our methods again.

# N  THE ROC CURVES OF DEFENSES

To better compare our method with baseline defenses, we also visualize the ROC curves of defenses (as shown in Figure 15-16) under each attack on both CIFAR-10 and Tiny ImageNet in this section.

# O  DETAILS FOR THE EFFECTIVENESS OF SCALING PROCESS

Specifically, we design the SCALE-UP variant by replacing the scaling process with adding the same varied magnitudes of random noise to the given inputs. As shown in Table 7, using random noises is far less effective compared to the standard SCALE-UP methods, especially in detecting advanced attacks (*i.e.*, WaNet and ISSBA). We speculate that it is mostly because they adopted invisible full-image size trigger patterns and therefore the trigger-related features are less robust. Although we currently fail to provide theoretical analysis for the aforementioned phenomena, at least they verify the effectiveness of our scaling process. We will further discuss it in our future work.

# P  POTENTIAL LIMITATIONS AND FUTURE WORK

Our work is the first black-box label-only input-level backdoor detection and early-stage defenses under the black-box setting. Accordingly, we have to admit that our work still has some limitations.

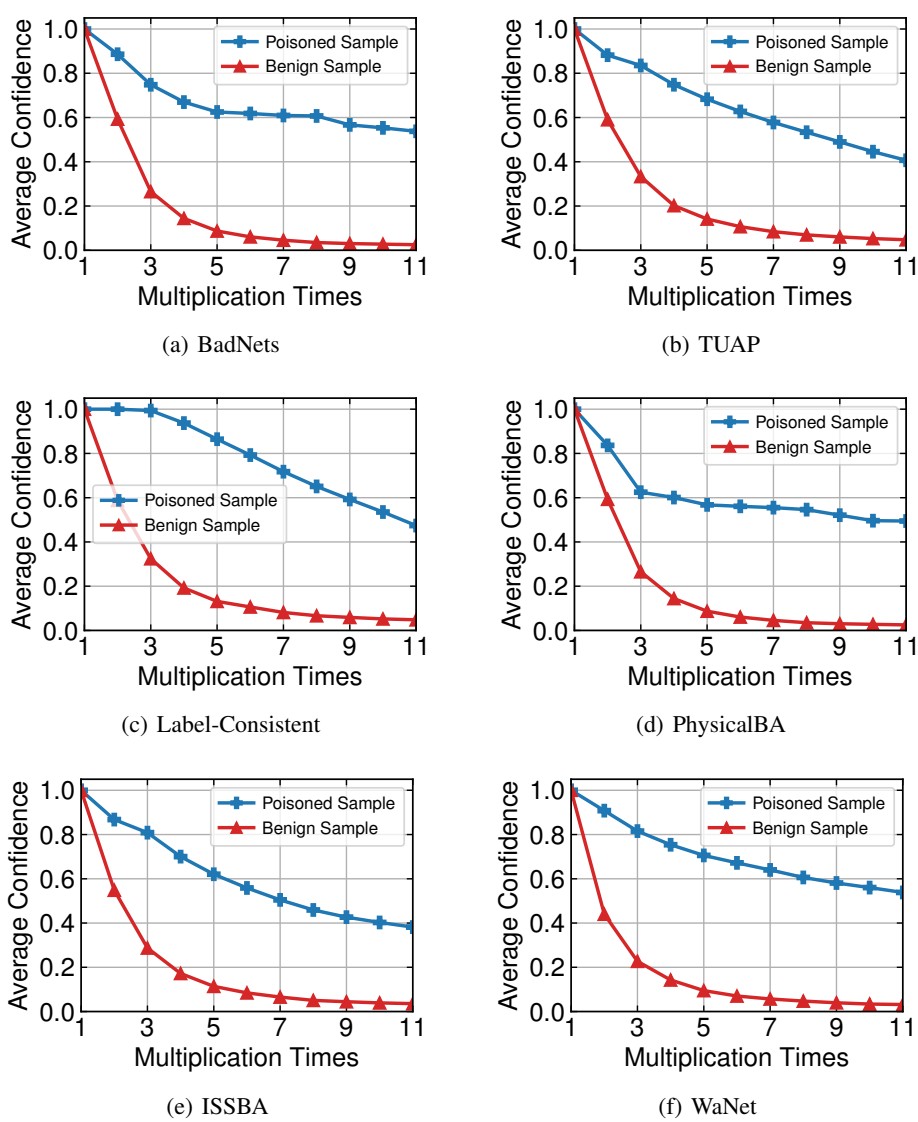

Figure 14: The average confidence (*i.e.*, average probabilities on the originally predicted label) of benign and poisoned samples *w.r.t.* pixel-wise multiplications under benign and attacked models on the Tiny ImageNet dataset with VGG-19 (with batch normalization).

Firstly, our defense requires that the attacked DNNs overfit their poisoned samples. This assumption or potential limitation is also revealed by our theoretical analysis in Section 3. In other words, if the attack success rate of a malicious model is relatively low, the detection performance of our SCALE-UP defense may degrade sharply. Secondly, we found that our SCALE-UP detection may fail when defending against attacks in some cases of simple tasks (*e.g.*, MNIST and GTSRB). We speculate that it is mostly because attacked DNNs also overfit to benign samples due to the lack of diversity and simplicity of the dataset, making them indistinguishable from some poisoned samples when analyzing the scaled prediction consistency. We will further explore the latent mechanisms of these limitations and alleviate them in our future work.

Besides, regarding another future direction of our methods, we intend to generalize and adopt them to more settings and applications, such as continual learning (Wang et al., 2022b), non-transferable learning (Wang et al., 2022a), federate learning (Dong et al., 2022), audio signal processing (Zhai et al., 2021; Guo et al., 2022a;b), and visual object tracking (Li et al., 2022b). We will also evaluate our methods under other DNN structures (*e.g.*, ViT (Tu et al., 2022) and GCN (Zhao et al., 2020b)).

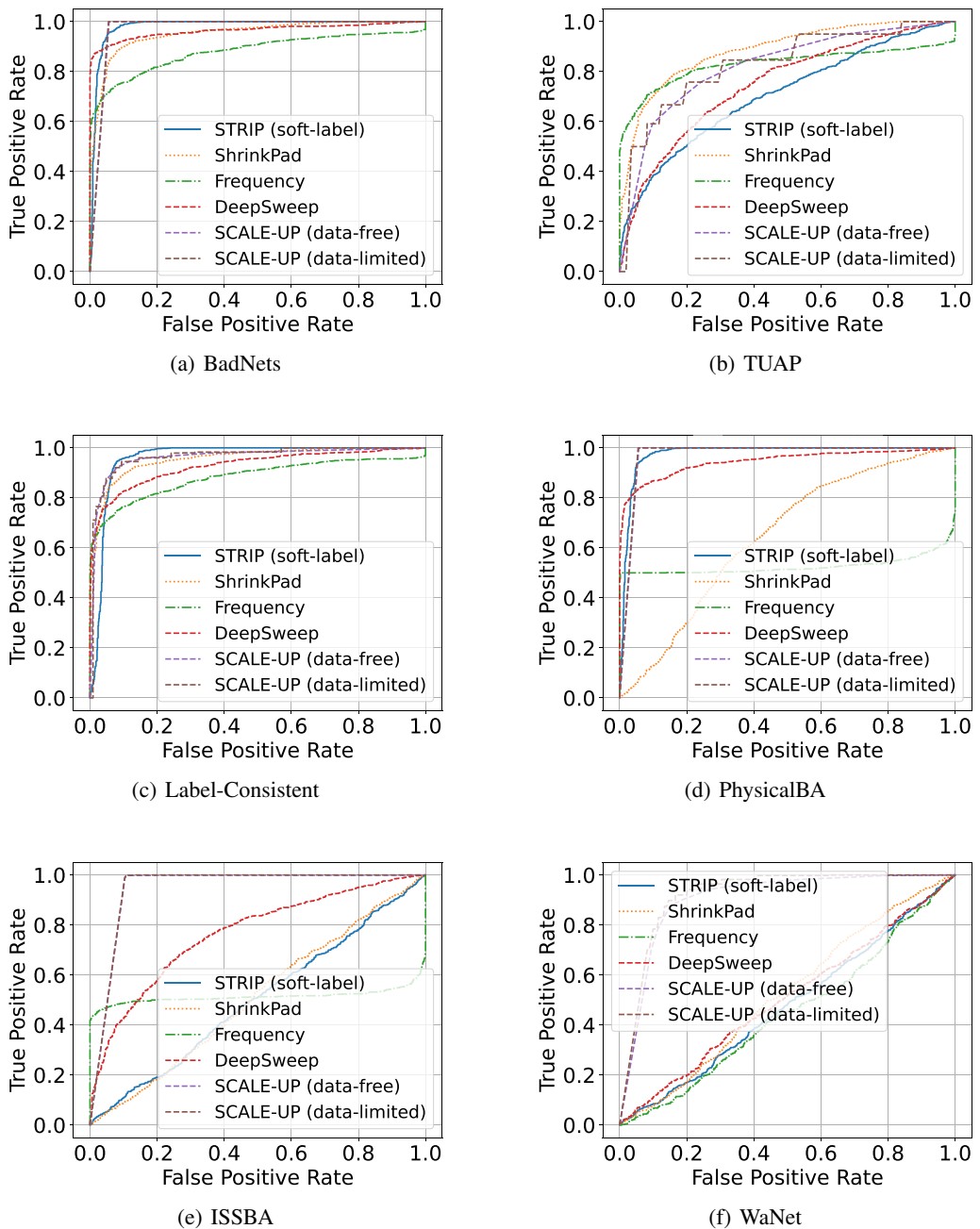

Figure 15: The ROC curves of defenses under each attack on CIFAR-10.

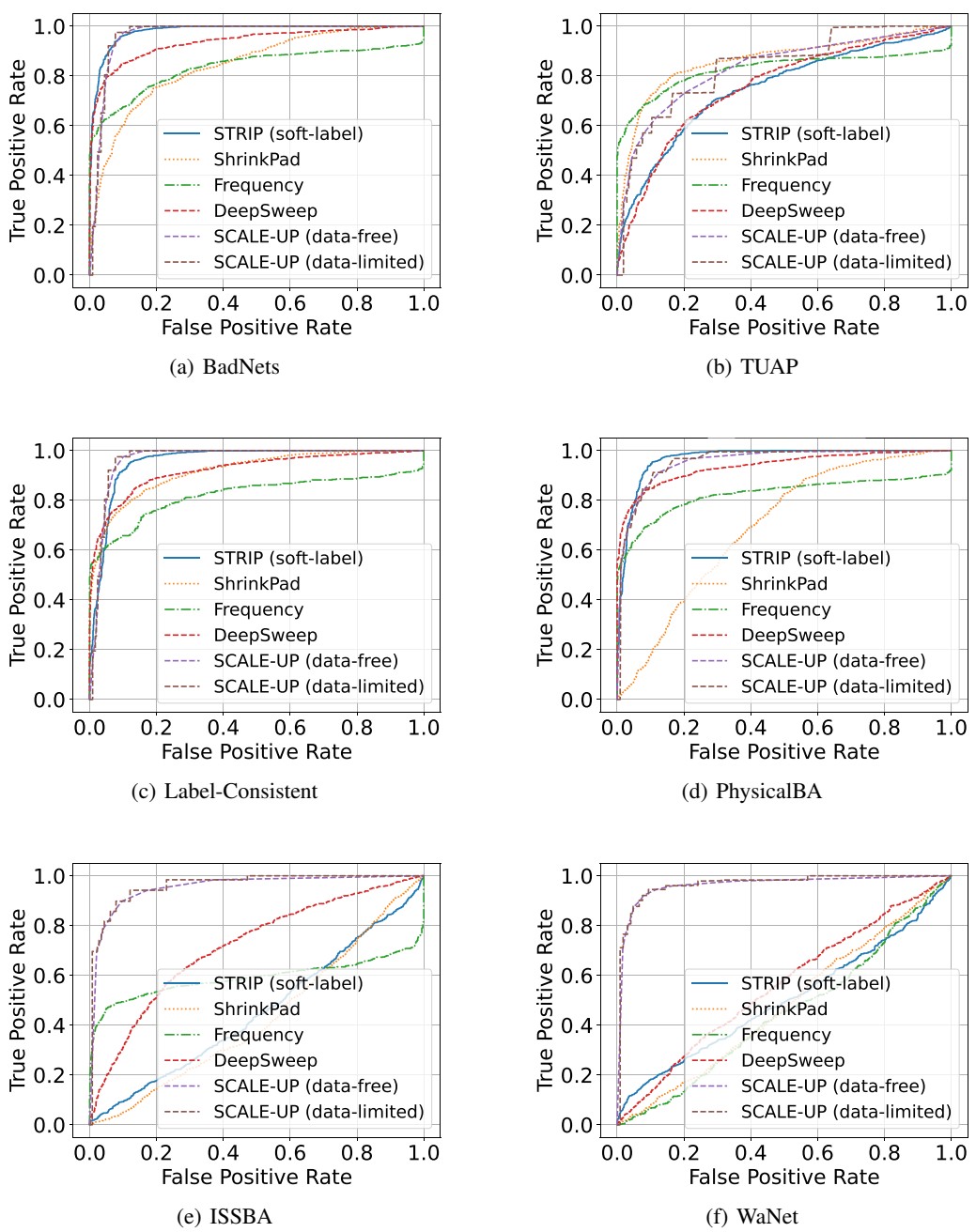

Figure 16: The ROC curves of defenses under each attack on Tiny ImageNet.

