# OpenReview forum: "SCALE-UP: An Efficient Black-box Input-level Backdoor Detection via Analyzing Scaled Prediction Consistency"
_ICLR.cc/2023/Conference — ICLR 2023 poster_

### Official Review · Reviewer_JRLs · 2022-10-21

**Confidence:** 3
**Correctness:** 4
**Technical Novelty And Significance:** 3
**Empirical Novelty And Significance:** 3
**Recommendation:** 8

**Clarity, Quality, Novelty And Reproducibility:**

#### **Clarity**: The paper is put very well together. The authors provide ample justifications for each design choice and try to convey that clearly to the readers.

#### **Quality**: The paper is of high quality. The observations made in the paper would indeed be interesting to many researchers in this area.

#### **Novelty And Reproducibility**: The proposed method is novel and interesting. The authors have also provided the code for their approach.

**Strength And Weaknesses:**

### Strengths:
- The paper is very well-written. The motivation behind each section is clear, and the paper navigates the reader smoothly.
- The observations around *scaled prediction consistency* shown in the paper are pretty interesting. The authors also take a step further and provide theoretical justification for the observed phenomenon using NTKs (though this reviewer hasn't checked the proof rigorously.)
- The experimental analysis of the proposed method seems thorough. More importantly, the paper presents various counter-arguments around the proposed method (such as adaptive attacks against the proposed defense and analysis of noisy input images) and provides a fair empirical analysis for each case.

### Weaknesses:
The most critical weakness of this work is the lack of diversity in terms of neural network architectures. To the best of this reviewer's attention, the method was only evaluated over ResNet. This choice would impose the question of whether the same observations are valid for different DNN architectures, such as various ConvNets and vision transformers.

**Summary Of The Paper:**

This paper presents a backdoor/poisoned data detection for deep neural networks (DNN) in the black-box setting (i.e., the defender only has access to the model's final output). The authors first argue that DNN users might rely on third-party providers to get their models. However, since those models might be proprietary, the users can only query the model and get their final prediction. As such, they investigate how one can detect a backdoor input without having access to the model weights. To this end, the authors empirically show that DNNs exhibit *scaled prediction consistency* for poisoned data: an input image $\boldsymbol{x}$ and its scaled version $n\cdot\boldsymbol{x}$ consistently result in similar predictions if $\boldsymbol{x}$ is poisoned. A similar result is theoretically proved for neural tangent kernels (NTK). Based on these findings, a backdoor data detection called *SCALE-UP* is proposed. This method can be used with or without having access to benign samples. Experimental results over two datasets (CIFAR-10 and TinyImageNet) plus six different backdoor attacks show the proposed method's success in detecting backdoor-poisoned data.

**Summary Of The Review:**

Based on my understanding presented above, the paper introduces an interesting phenomenon about backdoor attacks and uses this as a motivation for a defense mechanism. The experimental results are interesting, though they lack diversity in DNN architecture. Overall, I think the paper can provide good insights for the researchers in this area. Thus, I recommend acceptance at this stage. Still, I'd be keen to see my peers' take on the paper.

---

> ### Author Response · Authors · 2022-11-11
> **Author Response**
>
> We sincerely thank you for your valuable time and constructive comments. We are encouraged by your positive comments on our **paper quality**, **motivation**, **interesting phenomenon**, **theoretical foundation**, **technical novelty**, and **comprehensive and fair experiments**! We will alleviate your remaining concerns as follows.
>
> **Note**: All modified contents are marked in orange in our revision.
>
> ---
> **Q1**: The most critical weakness of this work is the lack of diversity in terms of neural network architectures. To the best of this reviewer's attention, the method was only evaluated over ResNet. This choice would impose the question of whether the same observations are valid for different DNN architectures.
>
> **R1**: Thank you for this insightful comment! We do understand your concern and agree that it is critical to ensure that the scaled prediction consistency is valid across different DNN architectures. To alleviate your concern, we conduct additional experiments of defenses under VGG-19 (with BN) on the Tiny ImageNet dataset, as follows:
>
>
> Table 1. The performance (AUROC) on the Tiny ImageNet dataset under VGG-19. Among all different methods, the best result is marked in boldface while the value with underline denotes the second-best result. The failed cases ($i.e.$, AUROC $<0.55$) are marked in red. Note that STRIP requires obtaining predicted probability vectors while other methods only need the predicted labels.
> |         Defense$\downarrow$, Attack$\rightarrow$        | BadNets | Label Consistent | PhysicalBA |  TUAP | WaNet | ISSBA | Average |
> |:-------------------:|:-------:|:----------------:|:----------:|:-----:|:-----:|:-----:|:-------:|
> |        STRIP        |  0.941  |       0.908      |    0.941   | 0.576 | <font color='red'>0.521</font> | <font color='red'>0.489</font> |  0.729  |
> |      ShrinkPad      |  0.857  |       0.919      |    0.631   | 0.831 | <font color='red'>0.499</font> | <font color='red'>0.490</font> |  0.705  |
> |      DeepSweep      |  0.939  |       0.907      |    0.921   | 0.744 | <font color='red'>0.511</font> | 0.711 |  0.788  |
> |      Frequency      |  0.864  |       0.859      |    0.864   | 0.827 | <font color='red'>0.428</font> | <font color='red'>0.540</font> |  0.730  |
> |   Ours (Data-free)  |  0.936  |       0.846      |    0.907   | 0.858 | 0.893 | 0.767 |  0.868  |
> | Ours (Data-limited) |  0.936  |       0.851      |    0.907   | 0.888 | 0.904 | 0.836 |  0.887  |
>
> **The aforementioned results verify the effectiveness of our defenses again**. Besides, we also visualize the average confidence ($i.e.$, average probabilities on the originally predicted label) of benign and poisoned samples $w.r.t.$ pixel-wise multiplications under benign and each attacked models (as shown in Figure 15) in our revision. Please find more details in Appendix N of our revision.
>
>
> ---

---

> ### Author Response · Authors · 2022-11-15
> **Thanks to Reviewer JRLs**
>
> We would like to thank you again for reviewing our work and the valuable feedback, and in particular for recognizing the strengths of our paper in terms of *paper quality*, *motivation*, *interesting phenomenon*, *theoretical foundation*, *technical novelty*, and *comprehensive and fair experiments*.
>
> Please kindly let us know if you have any additional questions or require further clarification of the effectiveness of our method across different DNN architectures. We are happy to address them before the rebuttal ends.

---

> ### Author Response · Authors · 2022-11-21
> **A Gentle Reminder of the Final Feedback**
>
> We would like to thank the reviewer for the helpful discussion during the first round of the review. We hope our response has adequately addressed your comments related to the effectiveness of our method across different DNN architectures. We take this as a great opportunity to improve our work and shall be grateful for any additional feedback you could give to us.

---

### Official Review · Reviewer_4qGQ · 2022-10-22

**Confidence:** 3
**Correctness:** 3
**Technical Novelty And Significance:** 3
**Empirical Novelty And Significance:** 2
**Recommendation:** 6

**Clarity, Quality, Novelty And Reproducibility:**

The paper is well-written and easy to understand. Please provide the code at the beginning of the rebuttal for reproducibility.

**Strength And Weaknesses:**

The authors consider backdoor detection under the black-box setting in machine learning as a service (MLaaS) applications. And the proposed method alleviates backdoor threats in this case.

Strength:
+ The MLaaS setting makes sense. In real-world applications, developers and users may directly exploit third-party pre-trained DNNs instead of training their new models. The practical application of this work is instructive.
+ The phenomenon, dubbed scaled prediction consistency, is very interesting. It is novel to utilize this phenomenon to defend against the backdoor attack.
+ The method is easy and effective.

Though the proposal shows an excellent performance in the experiments presented in this paper, I still have some concerns about the adequacy of the experiment.

Weaknesses / Questions
+ In Table 1-2, the defense methods(STRIP, ShrinkPad, DeepSweep, Frequency) are all designed for patch-based attacks. However, the no-patch-based attacks are used for detection, which may be more favorable for the method in the article, and the comparison may be unfair. I suggest the authors add experiments about no-patch-based defenses for comparison.


**Summary Of The Paper:**

This paper considers backdoor threats under the real-world machine learning as a service (MLaaS) setting where users can only query and obtain predictions of the deployed model. In this setting, the existing defenses fail to work because they assume that the suspicious models are transparent to users and can be modified. To reduce backdoor threats in the MLaaS setting, the paper proposes a simple yet effective black-box input-level backdoor detection called SCALE-UP, which requires only the predicted labels. Motivated by an intriguing observation, the proposal identify and filter malicious testing samples by analyzing their prediction consistency. The experiments verify the effectiveness and efficiency of the proposed defense method.

**Summary Of The Review:**

The paper proposes a simple method to alleviate backdoor attacks when users cannot access or modify suspicious models. The motivation is clear, and the method is effective.

---

> ### Author Response · Authors · 2022-11-11
> **Author Response**
>
> We sincerely thank you for your valuable time and constructive comments. We are encouraged by your positive comments on our **practical setting and application**, **interesting phenomenon**, **technical novelty**, and **high effectiveness**! We will alleviate your remaining concerns as follows.
>
> **Note**: All modified contents are marked in orange in our revision.
>
> ---
> **Q1**: In Table 1-2, the defense methods (STRIP, ShrinkPad, DeepSweep, Frequency) are all designed for patch-based attacks. However, the no-patch-based attacks are used for detection, which may be more favorable for the method in the article, and the comparison may be unfair. I suggest the authors add experiments about no-patch-based defenses for comparison.
>
> **R1**: Thank you for your comments and we do understand your concerns. We are deeply sorry that our submission may cause some misunderstandings to you that we would like to clarify at this rebuttal, as follows:
>
> - Firstly, **all baseline defenses are not designed only for patch-based attacks**. As shown in Table 1-2 in our main manuscript (or the following Table 1), all baseline defenses can successfully detect TUAP (to some extent), whose trigger patterns are image-size additive perturbations instead of some local patches. These defenses are less effective in defending against WaNet and ISSBA since their triggers are sample-specific instead of because their triggers are non-patch-based.
> - **We have already compared our method with almost all baseline methods that could be used under the setting of black-box input-level backdoor detection**. We would be grateful if you could provide the name of black-box input-level backdoor detections that you think we have missed. We are willing to compare our method with them before the rebuttal ends.
>
> **Note**: For your convenience, we have placed the important results related to TUAP contained in Table 1-2 of our submission, as follows:
>
> Table 1. The performance (AUROC) of baseline defenses in detecting TUAP on CIFAR-10 and Tiny ImageNet datasets.
> | Dataset$\downarrow$, Defense$\rightarrow$ | STRIP | ShrinkPad | DeepSweep | Frequency |
> |:----------------:|:-----:|:---------:|:---------:|:---------:|
> |     CIFAR-10     | 0.671 |   0.869   |   0.743   |   0.851   |
> |   Tiny ImageNet  | 0.638 |   0.866   |   0.759   |   0.837   |
>
>
>
> ---
> **Q2**: Please provide the code at the beginning of the rebuttal for reproducibility.
>
> **R2**: Thank you for this constructive suggestion and we do agree that providing source codes is important for reproducibility. However, it seems that there is a misunderstanding. We have already included our codes in supplementary materials along with our submission, as we mentioned in Reproducibility Statement. To avoid misunderstanding, we have highlighted this sentence in italics in our revision.
>
>
>
> ---

---

> ### Author Response · Authors · 2022-11-15
> **Thanks to Reviewer 4qGQ**
>
> We would like to thank you again for reviewing our work and the valuable feedback, and in particular for recognizing the strengths of our paper in terms of *practical setting and application*, *interesting phenomenon*, *technical novelty*, and *high effectiveness*.
>
> Please kindly let us know if you have any additional questions or require further clarification of *the comparison to no-patch-based defenses* and *source codes*. We are happy to address them before the rebuttal ends.

---

> ### Author Response · Authors · 2022-11-21
> **A Gentle Reminder of the Final Feedback**
>
> We would like to thank the reviewer for the helpful discussion during the first round of the review. We hope our response has adequately addressed your comments related to the comparison to no-patch-based defenses and our source codes. We take this as a great opportunity to improve our work and shall be grateful for any additional feedback you could give to us.

---

> ### Author Response · Authors · 2022-11-30
> **A Second Reminder of the Post-rebuttal Feedback**
>
> Dear Reviewer 4qGQ,
>
> We greatly appreciate your initial comments. We totally understand that you may be extremely busy at this time. But we still hope that you could have a quick look at our responses to your concerns. We appreciate any feedback you could give to us. We also hope that you could kindly update the rating if your questions have been addressed. We are also happy to answer any additional questions before the rebuttal ends.
>
> Best Regards,
>
> Paper 2131 Authors

---

> ### Author Response · Authors · 2022-12-03
> **The Third Warm Reminder of the Post-rebuttal Feedback**
>
> Dear Reviewer 4qGQ,
>
> We notice that all other reviewers have posted their post-rebuttal comments to our response while we still have not received any further information from you. We greatly appreciate your initial comments. We fully understand that you may be extremely busy at this time. But we still hope that you could have a quick look at our responses to your concerns. We appreciate any feedback you could give us. We also hope that you could kindly update the rating if your questions have been addressed. We are also happy to answer any additional questions before the rebuttal ends.
>
> Best Regards,
>
> Paper 2131 Authors

---

### Official Review · Reviewer_GMnc · 2022-10-25

**Confidence:** 4
**Correctness:** 4
**Technical Novelty And Significance:** 3
**Empirical Novelty And Significance:** 3
**Recommendation:** 8

**Clarity, Quality, Novelty And Reproducibility:**

Mentioned in the strength of the paper.


**Details Of Ethics Concerns:**

No ethics concerns.

**Strength And Weaknesses:**

B. Strength:
- The like the idea and the writing which are very clear and easy to understand.
- The method is simple and effective.
- The analysis and discussions have a lot of insights.
- The appendix covers a lot of missing details in the paper.

C. Weaknesses:
- I think the key limitation of the method is that SCALE-UP does not recover the trojan pattern. In other words, it cannot identify if the model is trojaned or not offline (e,g, https://www.ijcai.org/proceedings/2019/647). So even though your method achieves a very high AUROC score, it is not applicable in real-time applications such as a self-driving car; you cannot afford to lose ~2% of the real-time frames due to the false positive samples.

- I don't understand why your method is only ~5% slower compared to the "no defense" method. You need to infer the sample images multiple times (up to 14 times as given in Figure 12).  Also, there is no computation reuse between different inferences (due to input changes). Can you explain this issue?

Minor: AUROC is the only metric you applied throughout the paper (and appendix). Is it possible to provide the evaluation score of other metrics as well? Or maybe you can simply plot out some ROC curves.


**Summary Of The Paper:**

A. Paper summary

- This paper proposes an easy-to-understand blackbox trojan detection method. By leveraging the phenomenon called "scaled prediction consistency", the author suggests scaling up the input images and checking the confidence score. If the confidence drops, the input should be a benign sample; otherwise, it is poisoned. The evaluation result shows that the method is very effective. The paper also proposes a new adaptive attacking trojan to fully understand the limitation of the current detection method introduced in the paper.

**Summary Of The Review:**

Questions of the paper are given in the "weakness". I don't have too many questions because the appendix answered most of them.

Overall, the paper is good but still has room for improvement.

---

> ### Author Response · Authors · 2022-11-11
> **Author Response (Part I)**
>
> We sincerely thank you for your valuable time and constructive comments. We are encouraged by your positive comments on our **idea and paper writing**, **method effectiveness**, **extensive and insightful experiments**, and **comprehensive appendix**! We will alleviate your remaining concerns as follows.
>
> **Note**: All modified contents are marked in orange in our revision.
>
> ---
> **Q1**: I think the key limitation of the method is that SCALE-UP does not recover the trojan pattern. In other words, it cannot identify if the model is trojaned or not offline (e.g., https://www.ijcai.org/proceedings/2019/647). So even though your method achieves a very high AUROC score, it is not applicable in real-time applications such as a self-driving car. You cannot afford to lose ~2% of the real-time frames due to the false positive samples.
>
> **R1**: Thank you for this insightful comment and we do understand your concerns. We are deeply sorry that our submission may cause you some misunderstandings that we want to clarify at this rebuttal, as follows:
> - We admit that our method cannot recover trigger patterns. However, **it not necessarily means that our method cannot be used to detect trojans offline**. For example, when the training dataset (of the suspicious model) is avaiable, defenders can use our method to filter poisoned training samples. If our method finds a sufficiently large amount of poisoned samples, the suspicious model can be regarded as being trojaned.
> - Even if our method cannot detect trojans offline, **it is still practical in real-world applications since it can serve as the ‘firewall’ helping to block and trace back malicious samples in MLaaS scenarios** (as we mentioned in Related Work). We are deeply sorry that we failed to make it clear in our original submission. We have added more explanations at the beginning of the penultimate paragraph in Introduction of our revision.
> - **Recovering trigger patterns is very challenging under our input-level backdoor detection setting since defenders have limited capacities**. There is no existing defense that can fulfill it. Existing methods that can recover trigger patterns are either model-level [1-5] or under the white-box setting [6]. However, we do agree that it would be better if we can also recover trigger patterns. We will further explore how to extend our method to support this functionality in our future work.
> - We admit that our method may obtain some false-positive samples if defenders intend to recall all poisoned samples. However, **there is a trade-off between recall and precision for all detection-based methods**, unless the AUROC reaches 100\% (which is usually impossible). Users can adjust the threshold $T$ involved in our method to trade-off recall and precision based on their specific needs in real-world applications.
>
>
>
> 1. DeepInspect: A Black-box Trojan Detection and Mitigation Framework for Deep Neural Networks. IJCAI, 2019.
> 2. Neural Cleanse: Identifying and Mitigating
> Backdoor Attacks in Neural Networks. IEEE S&P, 2019.
> 3. AEVA: Black-box Backdoor Detection Using Adversarial Extreme Value Analysis. ICLR, 2022.
> 4. Backdoor Scanning for Deep Neural Networks through K-Arm Optimization. ICML, 2021.
> 5. ABS: Scanning Neural Networks for Back-doors by
> Artificial Brain Stimulation. CCS, 2019.
> 6. SentiNet: Detecting Localized Universal Attacks Against Deep Learning Systems. IEEE S&P Workshop, 2020.
>
> ---
> **Q2**: I don't understand why your method is only ~5% slower compared to the "no defense" method. You need to infer the sample images multiple times (up to 14 times as given in Figure 12). Also, there is no computation reuse between different inferences (due to input changes). Can you explain this issue?
>
>
> **R2**: Thank you for this insightful question! We are deeply sorry that our submission may cause you some misunderstandings that we want to clarify at this rebuttal, as follows:
>
> - **We calculated the inference time of our method by feeding all scaled variants of the suspicious image in a batch into the deployed model instead of predicting them one by one**. Accordingly, the inference time of our method is similar to that of No Defense, thanks to the high efficiency of matrix operations. Defenders can easily and efficiently obtain all its scaled variants of the suspicious image before feeding them into the deployed model.
> - **The aforementioned method is fair** since we adopted the same batch-based method to the inference time of all baseline methods ($e.g.$, STRIP and ShrinkPad).
> - Even under some restricted scenarios where defenders can only obtain the prediction of a single image at each time, they can still exploit parral computation to achieve similar inference time with the cost of more computation memories.
>
> To avoid misunderstandings, we have added more details in Appendix G of our revision. Sorry again for misleading you.
>
> ---

---

> > ### Author Response · Authors · 2022-11-11
> > **Author Response (Part II)**
> >
> >
> > ---
> > **Q3**: AUROC is the only metric you applied throughout the paper (and appendix). Is it possible to provide the evaluation score of other metrics as well? Or maybe you can simply plot out some ROC curves.
> >
> > **R3**: Thank you for this constructive suggestion! We have added the ROC curves of defenses under each attack in Appendix O of our revision.
> >
> > ---

---

> ### Author Response · Authors · 2022-11-15
> **Thanks to Reviewer GMnc**
>
> We would like to thank you again for reviewing our work and the valuable feedback, and in particular for recognizing the strengths of our paper in terms of *idea and paper writing*, *method effectiveness*, *extensive and insightful experiments*, and *comprehensive appendix*.
>
> Please kindly let us know if you have any additional questions or require further clarification of *the ability to recover trigger patterns* and *our computational efficiency*. We are happy to address them before the rebuttal ends.

---

> ### Author Response · Authors · 2022-11-21
> **A Gentle Reminder of the Final Feedback**
>
> We would like to thank the reviewer for the helpful discussion during the first round of the review. We hope our response has adequately addressed your comments related to our ability to recover trigger patterns and our computational efficiency. We take this as a great opportunity to improve our work and shall be grateful for any additional feedback you could give to us.

---

> ### Author Response · Authors · 2022-11-30
> **A Second Reminder of the Post-rebuttal Feedback**
>
> Dear Reviewer GMnc,
>
> We greatly appreciate your initial comments. We totally understand that you may be extremely busy at this time. But we still hope that you could have a quick look at our responses to your concerns. We appreciate any feedback you could give to us. We also hope that you could kindly update the rating if your questions have been addressed. We are also happy to answer any additional questions before the rebuttal ends.
>
> Best Regards,
>
> Paper 2131 Authors

---

### Decision · Program_Chairs · 2023-01-20

**Decision:**

Accept: poster

**Justification For Why Not Higher Score:**

According to my expertise and reviewing process, this paper should belong to an Accept with poster.

**Justification For Why Not Lower Score:**

According to my expertise and reviewing process, this paper should belong to an Accept with poster.

**Metareview: Summary, Strengths And Weaknesses:**

This paper focuses on an easy-to-understand blackbox trojan detection method called SCALE-UP. By exploiting the phenomenon called scaled prediction consistency, the author suggests scaling up the input images and checking the confidence score. The philosophy behind sounds quite interesting to me, namely, if the confidence decreases, the input should be a benign sample; otherwise, it is poisoned. In theoretical part, the authors take a step further and provide theoretical justification for the observed phenomenon using NTKs. In experimental part, the evaluation result shows that the method is effective. Moreover, the paper proposes a new adaptive attacking trojan to fully understand the limitation of the current detection method introduced in the paper.

The clarity and novelty are clearly above the bar of ICLR. While the reviewers had some concerns on the real-time limitation of SCALE-UP, the authors did a particularly good job in their rebuttal. Thus, all of us have agreed to accept this paper for publication! Please include the additional experimental results and further explanation in the next version.

**Note From Pc:**

if the above contains the word "oral" or "spotlight" please see: "oral" presentation means -> notable-top-5% and "spotlight" means -> notable-top-25%. As stated in our emails, we are disassociating presentation type from AC recommendations